# NEURAL CAPACITANCE: A NEW PERSPECTIVE OF NEURAL NETWORK SELECTION VIA EDGE DYNAMICS

## ABSTRACT

Efficient model selection for identifying a suitable pre-trained neural network to a downstream task is a fundamental yet challenging task in deep learning. Current practice requires expensive computational costs in model training for performance prediction. In this paper, we propose a novel framework for neural network selection by analyzing the governing dynamics over synaptic connections (edges) during training. Our framework is built on the fact that back-propagation during neural network training is equivalent to the dynamical evolution of synaptic connections. Therefore, a converged neural network is associated with an equilibrium state of a networked system composed of those edges. To this end, we construct a network mapping $\phi$, converting a neural network $G_A$ to a directed line graph $G_B$ that is defined on those edges in $G_A$. Next, we derive a *neural capacitance* metric $\beta_{\text{eff}}$ as a predictive measure universally capturing the generalization capability of $G_A$ on the downstream task using only a handful of early training results. We carried out extensive experiments using 17 popular pre-trained ImageNet models and five benchmark datasets, including CIFAR10, CIFAR100, SVHN, Fashion MNIST and Birds, to evaluate the fine-tuning performance of our framework. Our neural capacitance metric is shown to be a powerful indicator for model selection based only on early training results and is more efficient than state-of-the-art methods.

## 1 INTRODUCTION

Leveraging a pre-trained neural network (i.e., a source model) and fine-tuning it to solve a target task is a common and effective practice in deep learning, such as transfer learning. Transfer learning has been widely used to solve complex tasks in text and vision domains. In vision, models trained on ImageNet are leveraged to solve diverse tasks such as image classification and object detection. In text, language models that are trained on a large amount of public data comprising of books, Wikipedia etc are employed to solve tasks such as classification and language generation. Although such technique can achieve good performance on a target task, a fundamental yet challenging problem is how to select a suitable pre-trained model from a pool of candidates in an efficient manner. The naive solution of training each candidate fully with the target data can find the best pre-trained model but is infeasible due to considerable consumption on time and computation resources. This challenge motivates the need for an efficient predictive measure to capture the performance of a pre-trained model on the target task *based only on early training results* (e.g., predicting final model performance based on the statistics obtained from first few training epochs).

In order to implement an efficient neural network (NN) model selection, this paper proposes a novel framework to forecast the predictive ability of a model with its cumulative information in the early phase of NN training, as practised in learning curve prediction (Domhan et al., 2015; Chandrashekaran & Lane, 2017; Baker et al., 2017; Wistuba & Pedapati, 2020). Most prior work on learning curve prediction aims to capture the trajectory of learning curves with a regression function of models' validation accuracy. Some of the previous algorithms developed in this field require training data from *additional* learning curves to train the predictors (Chandrashekaran & Lane, 2017; Baker et al., 2017; Wistuba & Pedapati, 2020). On the other hand, our model does not require any such data. It solely relies on the NN architecture. Ranking models according to their final accuracy after fine-tuning is a lot more challenging as the learning curves are very similar to each other.

The entire NN training process involves iterative updates of the weights of synaptic connections, according to one particular optimization algorithm, e.g., gradient descent or stochastic gradient descent (SGD) (Bottou, 2012; LeCun et al., 2015). In essence, many factors contribute to impact how weights are updated, including the training data, the neural architecture, the loss function, and the optimization algorithm. Moreover, weights evolving during NN training in many aspects can be viewed as a discrete dynamical system. The perspective of viewing NN training as a dynamical system has been studied by the community (Mei et al., 2018; Chang et al., 2018; Banburski et al., 2019; Dogra, 2020; Tano et al., 2020; Dogra & Redman, 2020; Feng & Tu, 2021), and many attempted to make some theoretical explanation of the convergence rate and generalization error bounds. In this paper, we will provide the first attempt in exploring its power in neural model selection.

One limitation of current approaches is that they concentrated on the macroscopic and collective behavior of the system, but lacks a dedicated examination of the individual interactions between the trainable weights or synaptic connections, which are crucial in understanding of the dependency of these weights, and how they co-evolve during training. To fill the gap, we study the system from a microscopic perspective, build edge dynamics of synaptic connections from SGD in terms of differential equations, from which we build an associated network as well. The edge dynamics induced from SGD is nonlinear and highly coupling. It will be very challenging to solve, considering millions of weights in many convolutional neural networks (CNNs), e.g., 16M weights in MobileNet (Howard et al., 2017) and 528M in VGG16 (Simonyan & Zisserman, 2014). Gao et al. (2016) proposed a universal topological metric for the associated network to decouple the system. The metric will be used for model selection in our approach, and it is shown to be powerful in search of the best predictive model. We illustrate our proposed framework in Fig.1.

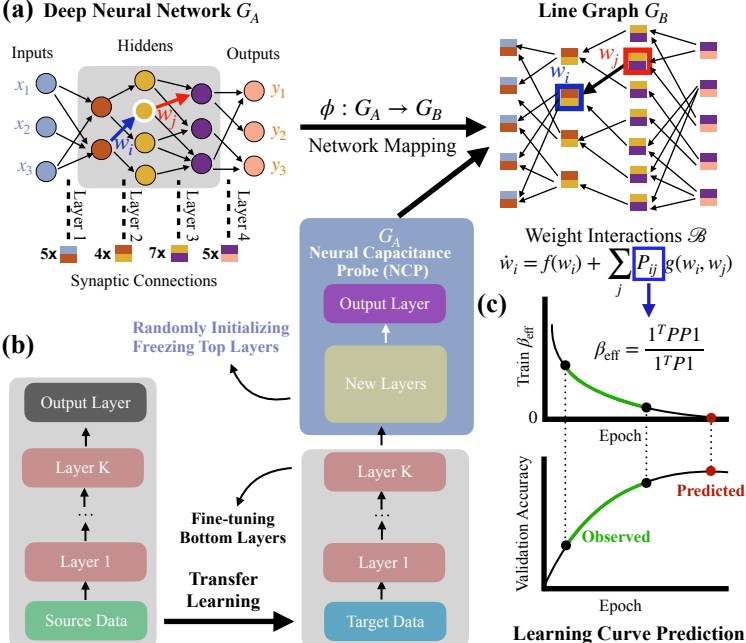

Figure 1: Illustration of our framework. **(a)** An example multilayer perceptron (MLP) $G_A$ is mapped to a directed line graph $G_B$, which is governed by an edge dynamics $\mathcal{B}$. Each node (dichromatic square) of $G_B$ is associated with a synaptic connection linking two neurons (in different colors) from different layers of $G_A$. **(b)** A diagram of transfer learning from the source domain (left stack) to a target domain (right stack). The pre-trained model is modified by adding additional layers, i.e. installing a neural capacitance probe (NCP) unit, on top of the bottom layers. The NCP is frozen with a set of randomly initialized weights, and only the bottom layers are fine-tuned. **(c)** Observed partial learning curves (green line segments) of validation accuracy over the early-stage training epochs and the corresponding neural capacitance metric $\beta_{\text{eff}}$ during fine-tuning. The predicted final accuracy at $\beta_{\text{eff}} \to 0$ (red dot) is used to select the best one from a set of models. The metric $\beta_{\text{eff}}$ relies on $G_B$'s weighted adjacency matrix $P$, which itself is derived from the reformulation of the training dynamics. To predict the performance, a lightweight $\beta_{\text{eff}}$ of the NCP is used instead of the heavyweight one over the entire network on the right stack of (b).

The main contributions of our framework can be summarized as follows:

- View NN training as a dynamical system over synaptic connections, and first time investigate the interactions of synaptic connections in a microscopic perspective.

- Propose neural capacitance metric $\beta_{\text{eff}}$ for neural network model selection.

- Empirical results of 17 pre-trained models on five benchmark datasets show that our $\beta_{\text{eff}}$ based approach outperforms current learning curve prediction approaches.

- For rank prediction according to the performance of pre-trained models, our approach improves by 9.1/38.3/12.4/65.3/40.1% on CIFAR10/CIFAR100/SVHN/Fashion MNIST/Birds over the best baseline with observations from learning curves of length only 5 epochs.

## 2 RELATED WORK

**Learning Curve Prediction.** Chandrashekaran & Lane (2017) treated the current learning curve (LC) as an affine transformation of previous LCs. They built an ensemble of transformations employing previous LCs and the first few epochs of the current LC to predict the final accuracy of the current LC. Baker et al. (2017) proposed an SVM based LC predictor using features extracted from previous LCs, including the architecture information such as number of layers, parameters, and training technique such as learning rate and learning rate decay. A separate SVM is used to predict the accuracy of an LC at a particular epoch. Domhan et al. (2015) trained an ensemble of parametric functions that observe the first few epochs of an LC and extrapolate it. Klein et al. (2017a) devised a Bayesian NN to model the functions that Domhan formulated to capture the structure of the LCs more effectively. Wistuba & Pedapati (2020) developed a transfer learning based predictor that was trained on LCs generated from other datasets. It is a NN based predictor that leverages architecture and dataset embeddings to capture the similarities between the architectures of various models and also the other datasets that it was trained on.

**Dynamical System View of NNs.** There are many efforts to study the dynamics of NN training. Some prior work on SGD dynamics for NNs generally have a pre-assumption of the input distribution or how the labels are generated. They obtained global convergence for shallow NNs (Tian, 2017; Banburski et al., 2019). System identification itself is a complicated task (Haykin, 2010; Lillicrap et al., 2020). In studying the generalisation phenomenon of deep NNs, Goldt et al. (2019) formulated SGD with a set of differential equations. But, it is limited to over-parameterised two-layer NNs under the teacher-student framework. The teacher network determines how the labels are generated. Also, some interesting phenomena (Frankle et al., 2020) are observed during the early phase of NN training, such as trainable sparse sub-networks emerge (Frankle et al., 2019), gradient descent moves into a small subspace (Gur-Ari et al., 2018), and there exists a critical effective connection between layers (Achille et al., 2019). Bhardwaj et al. (2021) built a nice connection between architectures (with concatenation-type skip connections) and the performance, and proposed a new topological metric to identify NNs with similar accuracy. Many of these studies are built on dynamical system and network science. It will be a promising direction to study deep learning mechanism.

## 3 PRELIMINARIES

**Dynamical System of a Network.** Many real complex systems, e.g., plant-pollinator interactions (Waser & Ollerton, 2006) and the spread of COVID-19 (Thurner et al., 2020), can be described with networks (Mitchell, 2006; Barabási & Pósfai, 2016). Let $G = (V, E)$ be a network with node set $V$ and edge set $E$. Assuming $n = |V|$, the interactions between nodes can be formulated as a set of differential equations

$$\dot{x}_i = f(x_i) + \sum_{j \in V} P_{ij} g(x_i, x_j), \forall i \in V, \tag{1}$$

where $x_i$ is the state of node $i$. In real systems, it could be the abundance of a plant in ecological network, the infection rate of a person in epidemic network, or the expression level of a gene in regulatory network. The term $P$ is the adjacency matrix of $G$, where the entry $P_{ij}$ indicates the interaction strength between nodes $i$ and $j$. The functions $f(\cdot)$ and $g(\cdot, \cdot)$ capture the internal and external impacts on node $i$, respectively. Usually, they are nonlinear.

Let $\boldsymbol{x} = (x_1, x_2, \ldots, x_n)$. For a small network, given an initial state, one can run a forward simulation for an equilibrium state $\boldsymbol{x}^*$, such that $\dot{x}_i^* = f(x_i^*) + \sum_{j \in V} P_{ij} g(x_i^*, x_j^*) = 0$. However, when the size of the system goes up to millions or even billions, it will pose a big challenge to solve the coupled differential equations. The problem can be efficiently addressed by employing a mean-field technique (Gao et al., 2016), where a linear operator $\mathcal{L}_P(\cdot)$ is introduced to decouple the system. In specific, the operator depends on the adjacency matrix $P$ and is defined as

$$\mathcal{L}_P(\boldsymbol{z}) = \frac{\mathbf{1}^T P \boldsymbol{z}}{\mathbf{1}^T P \mathbf{1}}, \tag{2}$$

where $\boldsymbol{z} \in \mathcal{R}^n$. Let $\boldsymbol{\delta}_{\text{in}} = P\mathbf{1}$ be nodes' in-degrees and $\boldsymbol{\delta}_{\text{out}} = \mathbf{1}^T P$ be nodes' out-degrees. For a weighted $G$, the degrees are weighted as well. Applying $\mathcal{L}_P(\cdot)$ to $\boldsymbol{\delta}_{\text{in}}$, it gives

$$\beta_{\text{eff}} = \mathcal{L}_P(\boldsymbol{\delta}_{\text{in}}) = \frac{\mathbf{1}^T P \boldsymbol{\delta}_{\text{in}}}{\mathbf{1}^T \boldsymbol{\delta}_{\text{in}}} = \frac{\boldsymbol{\delta}_{\text{out}}^T \boldsymbol{\delta}_{\text{in}}}{\mathbf{1}^T \boldsymbol{\delta}_{\text{in}}}, \tag{3}$$

which proves to be a powerful metric to measure the resilience of networks, and has been applied to make reliable inferences from incomplete networks (Jiang et al., 2020b;a). We use it to measure the predictive ability of a NN (see Section 4.3), whose training in essence is a dynamical system. For an overview of the related technique, the readers are referred to Appendix H.

**NN Training is a Dynamical System.** Conventionally, training a NN is a nonlinear optimization problem. Because of the hierarchical structure of NNs, the training procedure is implemented by two alternate procedures: forward-propagation (**FP**) and back-propagation (**BP**), as described in Fig.1(**a**). During FP, data goes through the input layer, hidden layers, up to the output layer, which produces the predictions of the input data. The differences between the outputs and the labels of the input data are used to define an objective function $\mathcal{C}$, a.k.a training error function. BP proceeds to minimize $\mathcal{C}$, in a reverse way as did in FP, by propagating the error from the output layer down to the input layer. The trainable weights of synaptic connections are updated accordingly.

Let $G_A$ be a NN, $\boldsymbol{w}$ be the flattened weight vector for $G_A$, and $\boldsymbol{z}$ be the set of activation values. As a whole, the training of $G_A$ can be described with two coupled dynamics: $\mathcal{A}$ on $G_A$, and $\mathcal{B}$ on $G_B$, where nodes in $G_A$ are neurons, and nodes in $G_B$ are the synaptic connections. The coupling relation arises from the strong inter-dependency between $\boldsymbol{z}$ and $\boldsymbol{w}$: the states $\boldsymbol{z}$ (activation values or activation gradients) of $G_A$ are the parameters of $\mathcal{B}$, and the states $\boldsymbol{w}$ of $G_B$ are the trainable parameters of $G_A$. If we put the whole training process in the context of networked systems, $\mathcal{A}$ denotes a *node dynamics* because the states of nodes evolve during FP, and $\mathcal{B}$ expresses an *edge dynamics* because of the updates of edge weights during BP (Mei et al., 2018; Poggio et al., 2020a;b). Mathematically, we formulate the node and edge dynamics based on the gradients of $\mathcal{C}$:

$$(\mathcal{A}) \quad d\boldsymbol{z}/dt \approx h_{\mathcal{A}}(\boldsymbol{z}, t; \boldsymbol{w}) = -\nabla_{\boldsymbol{z}} \mathcal{C}(\boldsymbol{z}(t)), \tag{4}$$

$$(\mathcal{B}) \quad d\boldsymbol{w}/dt \approx h_{\mathcal{B}}(\boldsymbol{w}, t; \boldsymbol{z}) = -\nabla_{\boldsymbol{w}} \mathcal{C}(\boldsymbol{w}(t)), \tag{5}$$

where $t$ denotes the training step. Let $a_i^{(\ell)}$ be the pre-activation of node $i$ on layer $\ell$, and $\sigma_\ell(\cdot)$ be the activation function of layer $\ell$. Usually, the output activation function is a softmax. The hierarchical structure of $G_A$ exerts some constraints over $\boldsymbol{z}$ for neighboring layers, i.e., $z_i^{(\ell)} = \sigma_\ell(a_i^{(\ell)}), 1 \leq i \leq n_\ell, \forall 1 \leq \ell < L$ and $z_k^{(L)} = \exp\{a_k^{(L)}\}/\sum_j \exp\{a_j^{(L)}\}, 1 \leq k \leq n_L$, where $n_\ell$ is the total number of neurons on layer $\ell$, and $G_A$ has $L + 1$ layers. It also presents a dependency between $\boldsymbol{z}$ and $\boldsymbol{w}$. For example, when $G_A$ is an MLP without bias, $a_i^{(\ell)} = \boldsymbol{w}_i^{(\ell)T} \boldsymbol{z}^{(\ell-1)}$, which builds an interconnection from $G_A$ to $G_B$. It is obvious, given $\boldsymbol{w}$, the activation $\boldsymbol{z}$ satisfying all these constraints, is also a fixed point of $\mathcal{A}$. Meanwhile, an equilibrium state of $\mathcal{B}$ provides a set of optimal weights for $G_A$.

## 4 OUR FRAMEWORK

The metric $\beta_{\text{eff}}$ is a universal metric to characterize different types of networks, including biological neural networks (Shu et al., 2021) (Section 3). Because of the generality of $\beta_{\text{eff}}$, we analyze how it looks on artificial neural networks which are designed to mimic the biological counterparts for general intelligence. Therefore, we set up an analogue system for the trainable weights. To the end, we build a line graph for the trainable weights (Section 4.1), and reformulate the training dynamics in the same form of the general dynamics (Eq. 1) (Section 4.2). The reformulated dynamics reveals

a simple yet powerful property regarding $\beta_{\text{eff}}$ (Section 4.3), which is utilized to predict the final accuracy of $G_A$ with a few observations during the early phase of the training (Section 4.4). For a detailed description of the core idea of our framework, see Appendix I.

## 4.1 LINE GRAPH $G_B$

We build a mapping scheme $\phi : G_A \mapsto G_B$, from an NN $G_A$ to an associated graph $G_B$. The topology of the synaptic connections (edges) is established as a well-defined line graph proposed by Nepusz & Vicsek (2012), and nodes of $G_B$ are the synaptic connections of $G_A$. More precisely, each node in $G_B$ is associated with a trainable parameter in $G_A$. For an MLP, each synaptic connection is assigned a trainable weight, the edge set of $G_A$ is also the set of synaptic connections of $G_B$. For a CNN, this one-to-one mapping from neurons on layer $\ell$ to layer $\ell + 1$ is replaced by a one-to-more mapping because of weight-sharing, e.g., a parameter in a convolutional filter is repeatedly used in FP and associated with multiple pairs of neurons from the two neighboring layers. Since the error gradients flow in a reversed direction, we reverse the corresponding links of the proposed line graph for $G_B$. In specific, given any pair of nodes in $G_B$, if they share an associated intersection neuron in FP propagation routes, a link with a reversed direction will be created for them. In Fig.1(**a**), we demonstrate how the mapping is performed on an example MLP. We have the topology of $G_B$ in place, but the weights of links in $G_B$ are not yet specified. To make up this missing components, we reveal the interactions of synaptic connections from SGD, quantify the interaction strengths and then define the weights of links in $G_B$ accordingly. Related technical details are disclosed in next section.

## 4.2 EDGE DYNAMICS $\mathcal{B}$

In SGD, each time a small batch of samples are chosen to update $\boldsymbol{w}$, i.e., $\boldsymbol{w} \leftarrow \boldsymbol{w} - \alpha \nabla_{\boldsymbol{w}} \mathcal{C}$, where $\alpha > 0$ is the learning rate. When desired conditions are met, training is terminated.

We denote the activation gradients as $\boldsymbol{\delta}^{(\ell)} = [\partial \mathcal{C}/\partial z_1^{(\ell)}, \cdots, \partial \mathcal{C}/\partial z_{n_\ell}^{(\ell)}]^T \in \mathcal{R}^{n_\ell 1}$ and the derivatives of activation function $\sigma$ for layer $\ell$ as $\boldsymbol{\sigma}'_\ell = [\sigma'_\ell(a_1^{(\ell)}), \cdots, \sigma'_\ell(a_{n_\ell}^{(\ell)})]^T \in \mathcal{R}^{n_\ell}, 1 \le \ell \le L$. To understand how the weights $W^{(\ell)}$ affect each other, we explicitly expand $\boldsymbol{\delta}^{(\ell)}$:

$$\boldsymbol{\delta}^{(\ell)} = W^{(\ell+1)T}(W^{(\ell+2)T}(\cdots(W^{(L-1)T}(W^{(L)T}(\boldsymbol{z}^{(L)} - \boldsymbol{y})) \odot \boldsymbol{\sigma}'_{L-1}) \cdots) \odot \boldsymbol{\sigma}'_{\ell+2}) \odot \boldsymbol{\sigma}'_{\ell+1}),$$

where $\odot$ is the Hadamard product. We find that parameters $W^{(\ell)}$ are associated with all accessible parameters on downstream layers, and such recursive relation defines a high-order hyper-network interaction (Casadiego et al., 2017) between any $W^{(\ell)}$ and the other parameters. The Hadamard product $\boldsymbol{x} \odot \boldsymbol{y}$ has an equivalent matrix multiplication form, i.e. $\boldsymbol{x} \odot \boldsymbol{y} = \Lambda(\boldsymbol{y})\boldsymbol{x}$, where $\Lambda(\boldsymbol{y})$ is a diagonal matrix consisting of the entries of $\boldsymbol{y}$ on the diagonal. Therefore, we have $\boldsymbol{\delta}^{(\ell)} = W^{(\ell+1)T}\Lambda(\boldsymbol{\sigma}'_{\ell+1})\boldsymbol{\delta}^{(\ell+1)}$ and $\boldsymbol{\delta}^{(\ell)} = W^{(\ell+1)T}\Lambda(\boldsymbol{\sigma}'_{\ell+1})W^{(\ell+2)T}\Lambda(\boldsymbol{\sigma}'_{\ell+2}) \cdots W^{(L-1)T}\Lambda(\boldsymbol{\sigma}'_{L-1})W^{(L)T}(\boldsymbol{z}^{(L)} - \boldsymbol{y})$. For a ReLU $\sigma_\ell(\cdot)$, $\boldsymbol{\sigma}'_\ell$ is binary depending on the sign of the input pre-activation values $\boldsymbol{a}^{(\ell)}$ of layer $\ell$. If $a_i^{(\ell)} \le 0$, then $\sigma'_\ell(a_i^{(\ell)}) = 0$, blocking a BP propagation route of the prediction deviations $\boldsymbol{z}^{(L)} - \boldsymbol{y}$ and giving rise to *vanishing gradients*.

Our purpose is to build direct interactions between synaptic connections. It can be done by identifying which units provide direct physical interactions to a given unit and appear on the right hand side of its differential equation $\mathcal{B}$ in Eq. 4, and how much such interactions come into play. There are multiple routes to build up a direct interaction between any pair of network weights from different layers, as presented by the product terms in $\boldsymbol{\delta}^{(\ell)}$. However, the coupled interaction makes it an impossible task, which is well known as a *credit assignment problem* (Whittington & Bogacz, 2019; Lillicrap et al., 2020). We propose a remedy. The impacts of all the other units on $W^{(\ell)}$ is approximated by direct, local impacts from $W^{(\ell+1)}$, and the others' contribution as a whole is implicitly encoded in the activation gradient $\delta^{(\ell+1)}$.

Moreover, we have the weight gradient (see Appendix A for detailed derivation)

$$\boldsymbol{\nabla}_{W^{(\ell)}} = \Lambda(\boldsymbol{\sigma}'_\ell)\boldsymbol{\delta}^{(\ell)}\boldsymbol{z}^{(\ell-1)T} = \Lambda(\boldsymbol{\sigma}'_\ell)W^{(\ell+1)T}\Lambda(\boldsymbol{\sigma}'_{\ell+1})\boldsymbol{\delta}^{(\ell+1)}\boldsymbol{z}^{(\ell-1)T}, \tag{6}$$

---

[1]In some literature $\boldsymbol{\delta}^{(\ell)}$ is defined as gradients with respect to $\boldsymbol{a}^{(\ell)}$, which does not affect our analysis.

which shows the dependency of $W^{(\ell)}$ on $W^{(\ell+1)}$, and itself can be viewed as an explicit description of the dynamical system $\mathcal{B}$ in Eq. 4. Put it in terms of a differential equation, we have

$$dW^{(\ell)}/dt = -\Lambda(\boldsymbol{\sigma}'_\ell)W^{(\ell+1)T}\Lambda(\boldsymbol{\sigma}'_{\ell+1})\boldsymbol{\delta}^{(\ell+1)}\boldsymbol{z}^{(\ell-1)T} \triangleq F(W^{(\ell+1)}). \tag{7}$$

Because of the mutual dependency of the weights and the activation values, it is hard to make an exact decomposition of the impacts of different parameters on $W^{(\ell)}$. However, in the gradient $\boldsymbol{\nabla}_{W^{(\ell)}}$, $W^{(\ell+1)}$ presents as an explicit term and contributes the direct impact on $W^{(\ell)}$. To capture such direct impact and derive the adjacency matrix $P$ for $G_B$, we apply Taylor expansion on $\boldsymbol{\nabla}_{W^{(\ell)}}$ and have

$$P^{(l,l+1)} = \partial^2 \mathcal{C}/\partial W^{(\ell)}\partial W^{(\ell+1)}, \tag{8}$$

which defines the interaction strength between each pair of weights from layer $\ell + 1$ to layer $\ell$. See Appendix B for detailed derivation of $P$ on MLP, and Appendix C on general NNs. Let $\boldsymbol{w} = (w_1, w_2, \ldots)$ be a flattened vector of all trainable weights of $G_A$. Given a pair of weights $w_i$ and $w_j$, one from layer $\ell_1$, another from layer $\ell_2$. If $\ell_2 = \ell_1 + 1$, the entry $P_{ij}$ is defined according to Eq. 8, otherwise $P_{ij} = 0$. Considering the scale of trainable parameters in $G_A$, $P$ is very sparse.

Let $W^{(\ell+1)*}$ be the equilibrium states (Appendix C), the training dynamics Eq. 7 is reformulated into the form of Eq. 1 and gives the edge dynamics $\mathcal{B}$ for $G_B$:

$$\dot{w}_i = f(w_i) + \sum_j P_{ij}g(w_i, w_j), \tag{9}$$

with $f(w_i) = F(w_i^*)$ and $g(w_i, w_j) = w_j - w_j^*$. The value of weights at an equilibrium state $\{w_j^*\}$ is unknown, but it is a constant and does not affect the computing of $\beta_{\text{eff}}$.

### 4.3 Neural Capacitance

According to Eq. 8, we have the weighted adjacency matrix $P$ of $G_B$ in place. Now we can quantify the total impact that a trainable parameter (or synaptic connection) receives from itself and the others, which corresponds to the weighted in-degrees $\boldsymbol{\delta}_{\text{in}} = P\mathbf{1}$. Applying $\mathcal{L}_P(\cdot)$ (see Eq. 2) to $\boldsymbol{\delta}_{\text{in}}$, we get a "counterpart" metric $\beta_{\text{eff}} = \mathcal{L}_P(\boldsymbol{\delta}_{\text{in}})$ to measure the predictive ability of a neural network $G_A$, as the resilience metric (see Eq. 3) does to a general network $G$ (see **Dynamical System of a Network** in Section 3). If $G_A$ is an MLP, we can explicitly write the entries of $P$, hence a $\beta_{\text{eff}}$ explicitly

$$\beta_{\text{eff}} = \frac{\sum_{\ell=2}^{L-2}[\mathbf{1}^T\boldsymbol{z}^{(\ell-2)}] \times \mathbf{1}^T[\boldsymbol{z}^{(\ell-1)} \odot \boldsymbol{\sigma}'_{\ell-1}] \times \mathbf{1}^T[\boldsymbol{\delta}^{(\ell)} \odot \boldsymbol{\sigma}'_\ell] \times \mathbf{1}^T[\boldsymbol{\delta}^{(\ell+1)} \odot \boldsymbol{\sigma}'_{\ell+1}]}{\sum_{\ell=2}^{L-1}[\mathbf{1}^T\boldsymbol{z}^{(\ell-2)}] \times [\mathbf{1}^T\boldsymbol{\sigma}'_{\ell-1}] \times \mathbf{1}^T[\boldsymbol{\delta}^{(\ell)} \odot \boldsymbol{\sigma}'_\ell]}. \tag{10}$$

For details of how to derive $P$ and $\beta_{\text{eff}}$ of an MLP, see Appendix B. Moreover, we prove in Theorem 1 below that as $G_A$ converges, $\boldsymbol{\nabla}_W^{(\ell)}$ vanishes, and $\beta_{\text{eff}}$ approaches zero (see Appendix D).

**Theorem 1.** *Let ReLU be the activation function of $G_A$. When $G_A$ converges, then $\beta_{\text{eff}} = 0$.*

For an MLP $G_A$, it is possible to derive an analytical form of $\beta_{\text{eff}}$. However, it becomes extremely complicated for a deep NN with multiple convolutional layers. To realize $\beta_{\text{eff}}$ for deep NNs in any form, we take advantage of the automatic differentiation implemented in TensorFlow[2]. Considering the number of parameters, it is still computationally expensive, and prohibitive to calculate a $\beta_{\text{eff}}$ for the entire $G_A$. Because of this, we seek to derive a surrogate from a partial of $G_A$. As shown in Section 4.4, we insert a *neural capacitance probe* (**NCP**) unit, i.e., putting additional lay-

---

**Algorithm 1** Implement NCP and Compute $\beta_{\text{eff}}$

**Input:** A pre-trained model $\mathcal{F}_s = \{\mathcal{F}_s^{(1)}, \mathcal{F}_s^{(2)}\}$ with bottom layers $\mathcal{F}_s^{(1)}$ and output layer $\mathcal{F}_s^{(2)}$, a target dataset $D_t$, the maximum number of epochs $T$

1: Remove $\mathcal{F}_s^{(2)}$ from $\mathcal{F}_s$ and add on top of $\mathcal{F}_s^{(1)}$ an NCP unit $\mathcal{U}$ with multiple layers (Fig.1**b**)
2: Initialize with random weights and freeze $\mathcal{U}$
3: Train $\mathcal{F}_t = \{\mathcal{F}_s^{(1)}, \mathcal{U}\}$ by fine-tuning $\mathcal{F}_s^{(1)}$ on $D_t$ for epochs of $T$
4: Obtain $P$ from $\mathcal{U}$ according to Eq. 8
5: Compute $\beta_{\text{eff}}$ with $P$ according to Eq. 3 or Eq. 10

---

ers on top of the beheaded $G_A$ (excluding the original output layer), and estimate the predictive ability of the entire $G_A$ using $\beta_{\text{eff}}$ of the NCP unit. Therefore, in the context of model selection from a pool of pre-trained models, if no confusion arises, we call $\beta_{\text{eff}}$ a *neural capacitance*.

---
[2]https://www.tensorflow.org/

### 4.4 Model Selection with $\beta_{\text{eff}}$

Here we show a novel application of our proposed neural capacitance $\beta_{\text{eff}}$ to model selection. In specific, we transfer the pre-trained models by (i) removing the output layer, (ii) adding some layers on top of the remaining layers (Fig.1**b**), and fine-tune them using a small learning rate. As shown in Algorithm 1, the newly added layers $\mathcal{U}$ on top of the bottom layers of $\mathcal{F}_s$ are used as an NCP unit. The specifics of the NCP unit are detailed in Section 5. The NCP does not involve in fine-tuning, and is merely used to calculate $\beta_{\text{eff}}$, then to estimate the performance of $G_A$ over the target domain $D_t$.

According to Theorem 1, when the model converges, $\beta_{\text{eff}} \to 0$. In an indirect way, the predictive ability of the model can be determined by the relation between the training $\beta_{\text{eff}}$ and the validation accuracy $I$. Since both $\beta_{\text{eff}}$ and $I$ are available during fine-tuning, we collect a set of data points of these two in the early phase as the observations, and fit a regularized linear model $I = h(\beta_{\text{eff}}; \boldsymbol{\theta})$ with Bayesian ridge regression (Tipping, 2001), where $\boldsymbol{\theta}$ are the associated coefficients (see Appendix E for technical details). The estimated predictor $I = h(\beta_{\text{eff}}; \theta^*)$ makes prediction of the final accuracy of models by setting $\beta_{\text{eff}} = 0$, i.e., $I^* = h(0; \theta^*)$, see an example in row 3 of Fig.2. For full training of the best model, one can either retain or remove the NCP and fine-tune the selected model.

## 5 Experiments and Results

**Pre-trained models and datasets.** We evaluate 17 pre-trained ImageNet models implemented in Keras[3], including AlexNet, VGGs (VGG16/19), ResNets (ResNet50/50V2/101/101V2/152/152V2), DenseNets (DenseNet121/169/201), MobileNets (MobileNet and MobileNetV2), Inceptions (InceptionV3, InceptionResNetV2) and Xception, to measure the performance of our approach. Four benchmark datasets CIFAR10, CIFAR100, SVHN, Fashion MNIST of size $32 \times 32 \times 3$, and one Kaggle challenge dataset Birds[4] of size $224 \times 224 \times 3$ are used, and their original train/test splits are adopted. In addition, 15K original training samples are set aside as validation set for each dataset.

**Experimental setup.** To get a well-defined $\beta_{\text{eff}}$, $G_A$ requires at least three hidden layers (see Appendix C). Also, a batch normalization (Ioffe & Szegedy, 2015) is usually beneficial because it can stabilize the training by adjusting the magnitude of activations and gradients. To this end, on top of each pre-trained model, we put a NCP unit composed of (1) a dense layer of size 256, (2) a dense layer of size 128, each of which follows (3) a batch normalization and is followed by (4) a dropout layer with a dropout probability of 0.4. Before fine-tuning, we initialize the NCP unit using Kaiming Normal initialization (He et al., 2015).

We set a batch size of 64 and a learning rate of 0.001, fine-tune each pre-trained model for $T = 50$ epochs, and repeated it for 20 times. As shown in Fig.2, the pre-trained models are converged after the fine-tuning on CIFAR10. For each model, we collect the validation accuracy (blue stars in row 1) and $\beta_{\text{eff}}$ on the training set (green squares in row 2) during the early stage of the fine-tuning as the observations (e.g., green squares in row 3 marked by the green box for 5 epochs), then use these observations to predict the test accuracy unseen before the fine-tuning terminates. For better illustration, learning curves are visualized on a log-scale.

**Evaluation.** We apply the Bayesian ridge regression on the observations to capture the relation between $\beta_{\text{eff}}$ and the validation accuracy, and to estimate a learning curve predictor $I = h(\beta_{\text{eff}}; \boldsymbol{\theta}^*)$. The performance of the model is revealed as $I^* = h(\beta_{\text{eff}}^*; \boldsymbol{\theta}^*)$ with $\beta_{\text{eff}}^* = 0$. As shown in row 3 of Fig.2, the blue lines are estimated $h(\cdot; \boldsymbol{\theta})$, the true test accuracy at $T$ and the predicted accuracy are marked as red triangles and blue stars, respectively. Both the estimates and predictions are accurate.

We aim to select the best one from a pool of candidates. A relative rank of these candidates matters more than their exact values of predicted accuracy. To evaluate and compare different approaches, we choose Spearman's rank correlation coefficient $\rho$ as the metric, and calculate $\rho$ over the true test accuracy at epoch $T$ and the predicted accuracy $I^*$ of all pre-trained models. In Fig.3(**a**), we report the true and predicted accuracy for each model on CIFAR10, as well as the overall ranking performance measured by $\rho$. It indicates that our $\beta$-based model ranking is reliable with $\rho > 0.9$. For the results on all five datasets, see Appendix Fig.F.4.

---

[3]https://keras.io/api/applications/

[4]https://www.kaggle.com/gpiosenka/100-bird-species

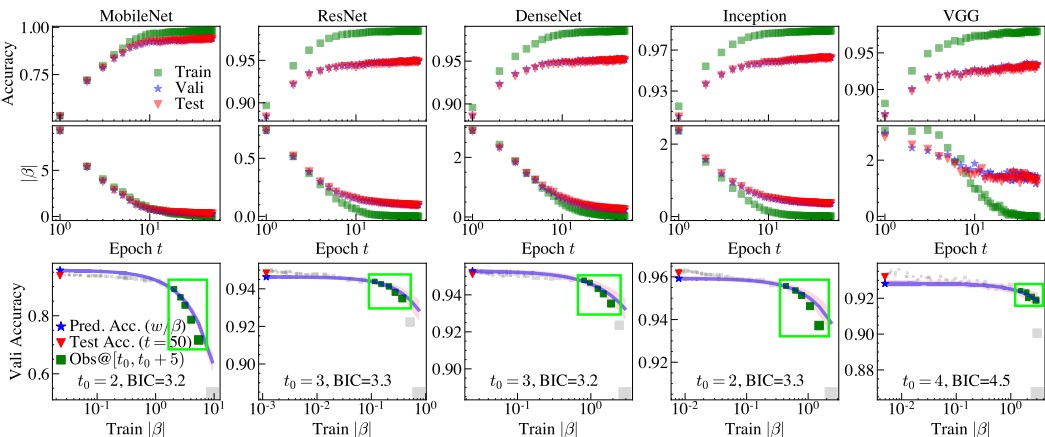

Figure 2: Learning curves of five representative pre-trained models w.r.t accuracy (row 1) and $\beta_{\text{eff}}$ (row 2). A regularized linear model $h(\cdot; \boldsymbol{\theta})$ (blue curve in row 3) is estimated with Bayesian ridge regression using a few of observations of $\beta_{\text{eff}}$ on training set and validation accuracy $I$ during early fine-tuning. The starting epoch $t_0$ of observations affects the fit of $h$, and is automatically determined according to BIC, and the true test accuracy at epoch 50 is predicted with $I^* = h(0; \boldsymbol{\theta}^*)$.

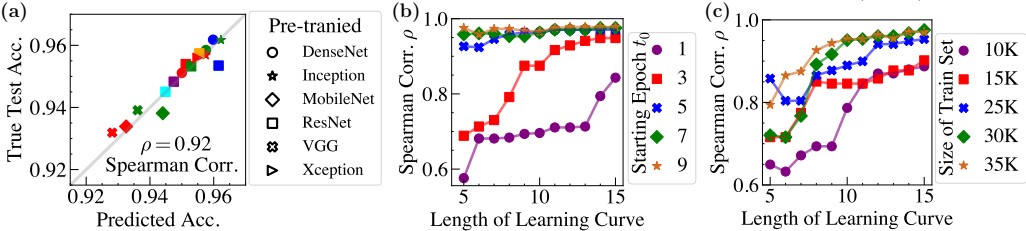

Figure 3: (a) Our $\beta_{\text{eff}}$ based prediction of the validation accuracy versus the true test accuracy at epoch 50 of seven representative pre-trained models. Each shape is associated with one type of pre-trained models. Distinct models of the same type are marked in different colors. Because the accuracy of AlexNet is much lower than others, we exclude it for better visualization. Its predicted accuracy is 0.871, and the true test accuracy is 0.868. If it is included, $\rho = 0.93 > 0.92$. (b) Impacts of the starting epoch $t_0$ of the observations and (c) the number of training samples on the ranking performance of our $\beta_{\text{eff}}$ based approach.

The estimation quality of $h$ determines how well the relation between $I$ and $\beta_{\text{eff}}$ is captured. Besides the regression method, the starting epoch $t_0$ of the observations also plays a role in the estimation. As shown in Fig.3(b), we evaluate the impact of $t_0$ on $\rho$ of our approach. It goes as expected, when the length of learning curves is fixed, a higher $t_0$ usually produces a better $\rho$. Since our ultimate goal is to predict with the early observations, $t_0$ should also be constrained to a small value. To make the comparisons fair, we view $t_0$ as a hyper-parameter, and select it according to the Bayesian information criterion (BIC) (Friedman et al., 2001), as shown in row 3 of Fig.2.

**Impact of size of training set.** CIFAR10 has 50K original training and 10K testing samples. Generally, the 50K samples are further split into 35K for training and 15K for validation. In studying the dynamics of the NN training, it is essential to understand how varying the training size influences the effectiveness of our approach. We select the first $\{10,15,20,25,30\}$K of the original 50K samples as the training set of reduced size, and the last 10K samples as the validation set to fine-tune the pre-trained models for 50 epochs. As shown in Fig.3(c), we can use a training set of size as small as 25K to achieve similar performance to that uses all 35K training samples. It has an important implication for efficient NN training, because the size of required training set can be greatly reduced (around 30% in our experiment) while maintaining similar model ranking performance. To be noted that the true test accuracy used in computing $\rho$ is the same test accuracy for the model trained from 35K training samples and it's shared by all the five cases $\{10,15,20,25,30\}$K in our analysis.

**Ours versus baselines.** We select BGRN (Baker et al., 2017) and CL (Chandrashekaran & Lane, 2017) as the baselines, as well as two heuristic rules of using the last seen value (LSV) (Klein et al., 2017b) or the best seen value (BSV) of a learning curve for extrapolation.

---

[6]https://github.com/tdomhan/pylearningcurvepredictor

Table 1: A comparison between our $\beta_{\text{eff}}$ based approach and the baselines in model ranking. The notation *LLC* represents the length of the learning curve, and *Imprv* represents the relative improvement of our approach to the best baseline. Due to the failure of the supporting package[6] of LC, there is a missing $\rho$ at *LLC* of 10, which does not affect our conclusions.

| Dataset | CIFAR10 | | CIFAR100 | | SVHN | | Fashion MNIST | | Birds | |
|---|---|---|---|---|---|---|---|---|---|---|
| *LLC* | 5 | 10 | 5 | 10 | 5 | 10 | 5 | 10 | 5 | 10 |
| Ours | **0.93** | **0.98** | **0.77** | 0.80 | **0.84** | **0.88** | **0.95** | **0.89** | **0.74** | **0.79** |
| BSV | 0.86 | 0.89 | 0.55 | 0.80 | 0.74 | 0.78 | 0.53 | 0.60 | 0.52 | 0.61 |
| LSV | 0.85 | 0.87 | 0.55 | 0.80 | 0.73 | 0.70 | 0.49 | 0.45 | 0.48 | 0.45 |
| BGRN | 0.74 | 0.78 | 0.45 | 0.60 | 0.63 | 0.65 | 0.57 | 0.59 | 0.53 | 0.52 |
| LC | 0.85 | 0.85 | 0.50 | 0.58 | 0.44 | 0.10 | 0.55 | 0.61 | 0.50 | – |
| *Imprv (%)* | 9.1 | 10.2 | 38.3 | -0.9 | 12.4 | 13.3 | 65.3 | 49.2 | 40.1 | 30.6 |

We compare the performance of ours with the baselines. As shown in Table 1 and Appendix Fig.F.5, using a few of observations, e.g., only 5 epochs, our approach can achieve 9.1/38.3/12.4/65.3/40.1% relative improvements over the best baseline on CIFAR10/CIFAR100/SVHN/Fashion MNIST/Birds.

**Running time analysis.** Our approach is efficient, especially for large and deep NNs. Different from the training task that involves a full FP and BP, i.e. $T_{\text{train}} = T_{\text{FP}} + T_{\text{BP}}$, computing $\beta_{\text{eff}}$ only requires to compute the adjacency matrix $P$ according to Eq. 8 on the NCP unit, $T_{\beta_{\text{eff}}} = T_{\text{NCP}}$. Although the computation is complicated, the NCP is lightweight. The computing cost per epoch is comparable to the training time per epoch (see Appendix Fig.G). Let $T_{\beta_{\text{eff}}} = c \times T_{\text{train}}$. If $c > 1$, i.e., $T_{\beta_{\text{eff}}}$ is higher than $T_{\text{train}}$, vice versa. Considering the required epochs, our approach needs $k$ observations, and takes $T_{\text{ours}} = k \times T_{\beta_{\text{eff}}}$. To obtain the ground-truth final accuracy by running $K$ epochs, it takes $T_{\text{full}} = K \times T_{\text{train}}$. If $T_{\text{full}} > T_{\text{ours}}$, our $\beta_{\text{eff}}$ based prediction is cheaper than "just training longer". It indicates that $K \times T_{\text{train}} - k \times T_{\beta_{\text{eff}}} = (K - c \times k) \times T_{\text{train}} > 0$, saving us $K - c \times k$ more training epochs.

We perform a running time analysis of the two tasks with $4\times$ NVIDIA Tesla V100 SXM2 32GB, and visualize the related times in Appendix Fig.G. On average $c = T_{\beta_{\text{eff}}}/T_{\text{train}} \approx 1.3$, computing $\beta_{\text{eff}}$ takes 1.3 times of the training per epoch. But the efforts are paying off, as we can predict the final accuracy by observing only $k = 10$ of $K = 100$ full training epochs, $T_{\text{ours}}$ is only 13% of $T_{\text{full}}$.

When the observations are used for learning curve prediction, the heuristics LSV and BSV directly take one observation (last or best) as the predicted value, so they are mostly computationally cheap but have suboptimal model ranking performances. Relatively, BGRN and CL are more time-consuming because both require to train a predictor with a set of full learning curves from other models. Our approach also estimates a predictor, but does not need any external learning curves. Here we assume that each model is observed for only $k = 5$ epochs, and conduct a running time analysis of these approaches over learning curve prediction, including estimating a predictor. As shown in Appendix Table G.2, our approach applies Bayesian ridge regression to efficiently estimate the predictor $I = h(\beta_{\text{eff}}; \boldsymbol{\theta})$, taking comparable time as BGRN, significantly less than CL, but performs best in model ranking. In contrast, the most expensive CL, does not perform well, sometimes even worst.

## 6 CONCLUSION AND DISCUSSION

We present a new perspective of NN model selection by directly exploring the dynamical evolution of synaptic connections during NN training. Our framework reformulates the SGD based NN training dynamics as an edge dynamics $\mathcal{B}$ to capture the mutual interaction and dependency of synaptic connections. Accordingly, a networked system is built by converting an NN $G_A$ to a line graph $G_B$ with the governing dynamics $\mathcal{B}$, which induces a definition of the link weights in $G_B$. Moreover, a topological property of $G_B$ named *neural capacitance* $\beta_{\text{eff}}$ is developed and shown to be an effective metric in predicting the ranking of a set of pre-trained models based on early training results.

There are several important directions that we intend to explore in the future, including (i) simplify the adjacency matrix $P$ to capture the dependency and mutual interaction between synaptic connections, e.g., approximate gradients using local information (Jaderberg et al., 2017), (ii) extend the proposed framework to NAS benchmarks (Ying et al., 2019; Dong & Yang, 2020; Dong et al., 2021; Zela et al., 2020; Li et al., 2021) to select the best subnetwork, and (iii) design an efficient algorithm to directly optimize NN architectures based on $\beta_{\text{eff}}$.

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

## A  ERROR GRADIENTS

Let $G_A$ be an MLP. To understand the learning mechanism, we take a sample $\boldsymbol{x}$ with label $\boldsymbol{y}$, and go through the entire training procedure, including a forward pass FP and a backward pass BP. To be convenient, we rewrite $\boldsymbol{z}^{(0)} = \boldsymbol{x}$ as the inputs, let $\boldsymbol{z}^{(1)}$ and $\boldsymbol{z}^{(L-1)}$ be the activations of the first and the last hidden layers, respectively, and let $\boldsymbol{z}^{(L)}$ be the outputs. The MLP is a parameterized model $\hat{\boldsymbol{y}} = \boldsymbol{z}^{(L)} = F_{\boldsymbol{w}}(\boldsymbol{x})$ with $\boldsymbol{w} = (W^{(1)}, W^{(2)}, \ldots, W^{(L)})$, where $W^{(\ell)}$ is the weight matrix of synaptic connections from layer $\ell - 1$ to layer $\ell$, and $1 \le \ell \le L$. Suppose there are $n_\ell$ neurons on layer $\ell$, $W^{(\ell)}$ has the size $n_\ell \times n_{\ell-1}$. The inputs $\boldsymbol{x}$ are fed into MLP, after a forward pass from layer 1 down to layer $L-1$ and layer $L$. Each neuron receives a cumulative input signal from the previous layer, and sends an activated signal to a downstream layer. Let $\sigma_\ell$ be the activation function of layer $\ell$, $a_i^{(\ell)} = \boldsymbol{w}_i^{(\ell)T} \boldsymbol{z}^{(\ell-1)}$ be the pre-activation value of neuron $i$ on layer $\ell$, we have $z_i^{(\ell)} = \sigma_\ell(a_i^{(\ell)})$ with $w_i^{(\ell)}$ being the $i$th row of $W^{(\ell)}$, $1 \le i \le n_\ell$. The output activation function $\sigma_L$ is generally a softmax function, i.e. $z_i^{(L)} = \exp\{a_i^{(L)}\}/\sum_j \exp\{a_j^{(L)}\}$, and $\boldsymbol{z}^{(L)}$ is a probability distribution over $n_L$ classes, i.e. $\mathbf{1}^T \boldsymbol{z}^{(L)} = 1$.

With the predictions $\boldsymbol{z}^{(L)}$ and the ground truth $\boldsymbol{y}$, we can calculate the prediction error $\mathcal{C}(\boldsymbol{z}^{(L)}, \boldsymbol{y})$, which is often a cross entropy loss, i.e. $\mathcal{C}(\boldsymbol{z}^{(L)}, \boldsymbol{y}) = -\sum_i y_i \log z_i^{(L)}$. To minimize $\mathcal{C}$, BP is applied, and the weights $\boldsymbol{w}$ are updated backward, from the output layer up to the first hidden layer.

Now we derive the gradients for a close examination. First, we get the derivatives of $\mathcal{C}$ w.r.t $\boldsymbol{z}^{(L)}$ and $W^{(L)}$. Because $z_i^{(L)} = \exp\{a_i^{(L)}\}/\sum_j \exp\{a_j^{(L)}\}$, we get the gradient of the output $z_k^{(L)}$ w.r.t $w_{ij}^{(L)}$:

$$\partial z_k^{(L)}/\partial w_{ij}^{(L)} = z_k^{(L)}(\delta_{ki} - z_i^{(L)})z_j^{(L-1)},$$

where $\delta_{ki} = 1$ if $k = i$, otherwise $\delta_{ki} = 0$.

On layer $L$, we get the derivatives of $\mathcal{C}$ w.r.t $\boldsymbol{z}^{(L)}$ and $W^{(L)}$:

$$\frac{\partial \mathcal{C}}{\partial z_i^{(L)}} = -\frac{y_i}{z_i^{(L)}}; \quad \frac{\partial \mathcal{C}}{\partial w_{ij}^{(L)}} = \sum_k \frac{\partial \mathcal{C}}{\partial z_k^{(L)}} \frac{\partial z_k^{(L)}}{\partial w_{ij}^{(L)}} = (z_i^{(L)} - y_i)z_j^{(L-1)} \tag{11}$$

Then, we examine layer $L-1$. The activation function $\sigma_{L-1}$ associates to a pair of neurons $u_i^{(L-1)}$ on layer $L-1$ and $u_j^{(L-2)}$ on layer $L-2$ with a unique connection weight $w_{ij}^{(L-1)}$. Since $z_i^{(L)} = \exp\{a_i^{(L)}\}/\sum_j \exp\{a_j^{(L)}\}$ and $z_i^{(L-1)} = \sigma_{L-1}(a_i^{(L-1)})$, we get $\partial z_k^{(L)}/\partial z_i^{(L-1)} = z_k^{(L)}(w_{ki}^{(L)} - \sum_j z_j^{(L)} w_{ji}^{(L)}) = z_k^{(L)}(w_{ki}^{(L)} - w_{*i}^{(L)T} \boldsymbol{z}^{(L)})$, and

$$\partial z_i^{(L-1)}/\partial w_{ij}^{(L-1)} = z_j^{(L-2)} \sigma'_{L-1}(a_i^{(L-1)}).$$

The derivatives of $\mathcal{C}$ are

$$\begin{aligned}
\frac{\partial \mathcal{C}}{\partial z_i^{(L-1)}} &= \sum_k \frac{\partial \mathcal{C}}{\partial z_k^{(L)}} \frac{\partial z_k^{(L)}}{\partial z_i^{(L-1)}} = \sum_k y_k(w_{*i}^{(L)T} \boldsymbol{z}^{(L)} - w_{ki}^{(L)}) = w_{*i}^{(L)T}(\boldsymbol{z}^{(L)} - \boldsymbol{y}), \\
\frac{\partial \mathcal{C}}{\partial w_{ij}^{(L-1)}} &= \frac{\partial \mathcal{C}}{\partial z_i^{(L-1)}} \frac{\partial z_i^{(L-1)}}{\partial w_{ij}^{(L-1)}} = w_{*i}^{(L)T}(\boldsymbol{z}^{(L)} - \boldsymbol{y})z_j^{(L-2)} \sigma'_{L-1}(a_i^{(L-1)}).
\end{aligned}$$

On layer $\ell$, where $1 \le \ell \le L-2$, we get $\partial z_k^{(\ell+1)}/\partial z_i^{(\ell)} = w_{ki}^{(\ell+1)} \sigma'_{\ell+1}(a_k^{(\ell+1)})$ and

$$\begin{aligned}
\frac{\partial \mathcal{C}}{\partial z_i^{(\ell)}} &= \sum_k \frac{\partial \mathcal{C}}{\partial z_k^{(\ell+1)}} \frac{\partial z_k^{(\ell+1)}}{\partial z_i^{(\ell)}} = \sum_k \frac{\partial \mathcal{C}}{\partial z_k^{(\ell+1)}} w_{ki}^{(\ell+1)} \sigma'_{\ell+1}(a_k^{(\ell+1)}), \\
\frac{\partial \mathcal{C}}{\partial w_{ij}^{(\ell)}} &= \frac{\partial \mathcal{C}}{\partial z_i^{(\ell)}} \frac{\partial z_i^{(\ell)}}{\partial w_{ij}^{(\ell)}} = \frac{\partial \mathcal{C}}{\partial z_i^{(\ell)}} z_j^{(\ell-1)} \sigma'_\ell(a_i^{(\ell)}),
\end{aligned}$$

according to the relations $z_i^{(\ell+1)} = \sigma_{\ell+1}(a_i^{(\ell+1)})$ and $z_i^{(\ell)} = \sigma_\ell(a_i^{(\ell)})$.

Let $\boldsymbol{\delta}^{(\ell)} = [\partial \mathcal{C}/\partial z_1^{(\ell)}, \cdots, \partial \mathcal{C}/\partial z_{n_\ell}^{(\ell)}]^T \in \mathcal{R}^{n_\ell}$, $\boldsymbol{\sigma}'_\ell = [\sigma'_\ell(a_1^{(\ell)}), \cdots, \sigma'_\ell(a_{n_\ell}^{(\ell)})]^T \in \mathcal{R}^{n_\ell}$, $1 \le \ell \le L$. The gradients can be written in a dense form:

$$\begin{aligned}
\nabla_{W^{(L)}} &= (\boldsymbol{z}^{(L)} - \boldsymbol{y})\boldsymbol{z}^{(L-1)T} \in \mathcal{R}^{n_L \times n_{L-1}}, \\
\boldsymbol{\delta}^{(L-1)} &= W^{(L)T}(\boldsymbol{z}^{(L)} - \boldsymbol{y}) \in \mathcal{R}^{n_{L-1}}, \\
\nabla_{W^{(L-1)}} &= (\boldsymbol{\delta}^{(L-1)} \odot \boldsymbol{\sigma}'_{L-1})\boldsymbol{z}^{(L-2)T} \in \mathcal{R}^{n_{L-1} \times n_{L-2}}, \\
\boldsymbol{\delta}^{(\ell)} &= W^{(\ell+1)T}(\boldsymbol{\delta}^{(\ell+1)} \odot \boldsymbol{\sigma}'_{\ell+1}) \in \mathcal{R}^{n_\ell}, \\
\nabla_{W^{(\ell)}} &= (\boldsymbol{\delta}^{(\ell)} \odot \boldsymbol{\sigma}'_\ell)\boldsymbol{z}^{(\ell-1)T} \in \mathcal{R}^{n_\ell \times n_{\ell-1}}.
\end{aligned} \tag{12}$$

## B   WEIGHTED DEGREES OF $G_B$

We examine three hidden layers $\{\ell - 1, \ell, \ell + 1\}$ of $G_A$ and three neurons on these layers $\{j, i, k\}$. Let $w_{ki}^{(\ell+1)}$ connects the neuron $k$ on layer $\ell + 1$ to the neuron $i$ on layer $\ell$, $w_{ij}^{(\ell)}$ be the synaptic connection weight between the neuron $j$ on layer $\ell$ and the neuron $i$ on layer $\ell - 1$, and $w_{jm}^{(\ell-1)}$ connects the neuron $j$ on layer $\ell - 1$ and the neuron $m$ on layer $\ell - 2$.

Now we have a close look at $\partial \mathcal{C} / \partial w_{ij}^{(\ell)}$. According to the chain rule, we have

$$\frac{\partial \mathcal{C}}{\partial w_{ij}^{(\ell)}} = \frac{\partial \mathcal{C}}{\partial z_i^{(\ell)}} \frac{\partial z_i^{(\ell)}}{\partial w_{ij}^{(\ell)}} = \delta_i^{(\ell)} z_j^{(\ell-1)} \sigma'_\ell(a_i^{(\ell)}) = z_j^{(\ell-1)} \sigma'_\ell(a_i^{(\ell)}) \sum_k \delta_k^{(\ell+1)} \sigma'_{\ell+1}(a_k^{(\ell+1)}) w_{ki}^{(\ell+1)}.$$

The gradient term $\delta_k^{(\ell+1)} = \partial \mathcal{C} / \partial z_k^{(\ell+1)}$ is a highly coupled function of all accessible synaptic connection weights of $w_{ij}^{(\ell)}$ on the forward propagation route from $z_i^{(\ell)}$ to the output neurons. To ease the analysis, we simplify it with a numerical value or a synthetic one with no synaptic connection weight included. Therefore, the summation term can be viewed as a simple linear function of all synaptic connection weights $w_{ki}^{(\ell+1)}$ associated with neuron $i$ on layer $\ell$, and the associated coefficient is $p\{w_{ki}^{(\ell+1)}, w_{ij}^{(\ell)}\} = z_j^{(\ell-1)} \sigma'_\ell(a_i^{(\ell)}) \delta_k^{(\ell+1)} \sigma'_{\ell+1}(a_k^{(\ell+1)})$, which defines the edge weights from $w_{ki}^{(\ell+1)}$ to $w_{ij}^{(\ell)}$ on $G_B$. Similarly, we have the edge weight from $w_{ij}^{(\ell)}$ to $w_{jm}^{(\ell-1)}$, i.e., $p\{w_{ij}^{(\ell)}, w_{jm}^{(\ell-1)}\} = z_m^{(\ell-2)} \sigma'_{\ell-1}(a_j^{(\ell-1)}) \delta_i^{(\ell)} \sigma'_\ell(a_i^{(\ell)})$. Therefore, we are able to calculate the in-degree and out-degree of $w_{ij}^{(\ell)}$, which are defined as the sum of the weights of all in-bound connections to $w_{ij}^{(\ell)}$ and the sum of the weights of all out-bound connections from $w_{ij}^{(\ell)}$, i.e.

$$\delta_{\text{in}}(w_{ij}^{(\ell)}) = z_j^{(\ell-1)} \sigma'_\ell(a_i^{(\ell)}) \Big[ \sum_k \delta_k^{(\ell+1)} \sigma'_{\ell+1}(a_k^{(\ell+1)}) \Big], \tag{13}$$

$$\delta_{\text{out}}(w_{ij}^{(\ell)}) = \Big[ \sum_m z_m^{(\ell-2)} \Big] \sigma'_{\ell-1}(a_j^{(\ell-1)}) \delta_i^{(\ell)} \sigma'_\ell(a_i^{(\ell)}). \tag{14}$$

There are several exceptions, including the first hidden ($\ell = 1$), the last hidden ($\ell = L - 1$) and the output ($\ell = L$) layers. For the output layer, we have

$$\frac{\partial \mathcal{C}}{\partial w_{ij}^{(L)}} = \sum_k \frac{\partial \mathcal{C}}{\partial z_k^{(L)}} \frac{\partial z_k^{(L)}}{\partial w_{ij}^{(L)}} = z_j^{(L-1)} (z_i^{(L)} - y_i). \tag{15}$$

Because $\sigma_L$ is softmax, no explicit relation regarding $w_{ij}^{(L)}$ can be built. It implies that no well-defined in-bound connections to $w_{ij}^{(L)}$, i.e., $\delta_{\text{in}}(w_{ij}^{(L)}) = 0$. But, we can build the connections from $w_{ij}^{(L)}$ to $w_{jm}^{(L-1)}$. It is easy to derive

$$\frac{\partial \mathcal{C}}{\partial w_{ij}^{(L-1)}} = \frac{\partial \mathcal{C}}{\partial z_i^{(L-1)}} \frac{\partial z_i^{(L-1)}}{\partial w_{ij}^{(L-1)}} = z_j^{(L-2)} \sigma'_{L-1}(a_i^{(L-1)}) \sum_k (z_k^{(L)} - y_k) w_{ki}^{(L)}. \tag{16}$$

From the perspective of $w_{ij}^{(L-1)}$, we get $p\{w_{ki}^{(L)}, w_{ij}^{(L-1)}\} = z_j^{(L-2)} \sigma'_{L-1}(a_i^{(L-1)})(z_k^{(L)} - y_k)$; from the perspective of $w_{ij}^{(L)}$, we have $p\{w_{ij}^{(L)}, w_{jm}^{(L-1)}\} = z_m^{(L-2)} \sigma'_{L-1}(a_j^{(L-1)})(z_i^{(L)} - y_i)$. So we have

$$\begin{aligned}
\delta_{\text{out}}(w_{ij}^{(L)}) &= \Big[ \sum_m z_m^{(L-2)} \Big] \sigma'_{L-1}(a_j^{(L-1)})(z_i^{(L)} - y_i), \\
\delta_{\text{in}}(w_{ij}^{(L-1)}) &= z_j^{(L-2)} \sigma'_{L-1}(a_i^{(L-1)}) \sum_k (z_k^{(L)} - y_k) = 0, \\
\delta_{\text{out}}(w_{ij}^{(L-1)}) &= \Big[ \sum_m z_m^{(L-3)} \Big] \sigma'_{L-2}(a_j^{(L-2)}) \delta_i^{(L-1)} \sigma'_{L-1}(a_i^{(L-1)}).
\end{aligned}$$

The softmax $\sigma_L(\cdot)$ makes the output values sum up to one, i.e., $\sum_k y_k = 1$, and $\delta_{\text{in}}(w_{ij}^{(L-1)}) = 0$. Now, we examine the first hidden layer. Similar to the output layer, there is no well-defined out-bound connections for $w_{ij}^{(1)}$, $\delta_{\text{out}}(w_{ij}^{(1)}) = 0$. Setting $\ell = 1$ in Eq. 13, we can get the in-degree of $w_{ij}^{(1)}$

$$\delta_{\text{in}}(w_{ij}^{(1)}) = z_j^{(0)} \sigma'_1(a_i^{(1)}) \Big[ \sum_k \delta_k^{(2)} \sigma'_2(a_k^{(2)}) \Big].$$

Based on our definition of the weights of $G_B$, when the number of layers is small, it is trivial that $\beta_{\text{eff}} = 0$. To get a non-trivial $\beta_{\text{eff}}$, we identify the minimum number of hidden layers in $G_A$. First, we examine a $G_A$ with one hidden layer, i.e. $L = 2$, whose degrees are

$$
\begin{aligned}
\delta_{\text{in}}(w_{ij}^{(1)}) &= \delta_{\text{out}}(w_{ij}^{(1)}) = \delta_{\text{in}}(w_{ij}^{(2)}) = 0, \\
\delta_{\text{out}}(w_{ij}^{(2)}) &= \big[\textstyle\sum_m z_m^{(0)}\big]\sigma_1'(a_j^{(1)})(z_i^{(2)} - y_i).
\end{aligned}
$$

Since the degrees sum up to zero, $\beta_{\text{eff}} = 0$, regardless of how many hidden neurons in $G_A$.

If $G_A$ only has two hidden layers, i.e. $L = 3$, the in-degrees are

$$
\begin{aligned}
\delta_{\text{in}}(w_{ij}^{(1)}) &= z_j^{(0)}\sigma_1'(a_i^{(1)})\big[\textstyle\sum_{k=1}^{n_2}\delta_k^{(2)}\sigma_2'(a_k^{(2)})\big], \\
\delta_{\text{in}}(w_{ij}^{(2)}) &= \delta_{\text{in}}(w_{ij}^{(3)}) = 0.
\end{aligned}
$$

The out-degrees are summarized as follows:

$$
\begin{aligned}
\delta_{\text{out}}(w_{ij}^{(1)}) &= 0, \\
\delta_{\text{out}}(w_{ij}^{(2)}) &= \big[\textstyle\sum_{m=1}^{n_0} z_m^{(0)}\big]\sigma_1'(a_j^{(1)})\delta_i^{(2)}\sigma_2'(a_i^{(2)}), \\
\delta_{\text{out}}(w_{ij}^{(3)}) &= \big[\textstyle\sum_{m=1}^{n_1} z_m^{(1)}\big]\sigma_2'(a_j^{(2)})(z_i^{(3)} - y_i).
\end{aligned}
$$

The total degree may be non-zero, but $\beta_{\text{eff}} = 0$ alway holds.

**Therefore, the minimum number of hidden layers required for a well-defined $\beta_{\text{eff}}$ is three, i.e., $L \geq 4$.** We summarize the in-degrees

$$
\begin{aligned}
\delta_{\text{in}}(w_{ij}^{(1)}) &= z_j^{(0)}\sigma_1'(a_i^{(1)})\big[\textstyle\sum_k\delta_k^{(2)}\sigma_2'(a_k^{(2)})\big], \\
\delta_{\text{in}}(w_{ij}^{(\ell)}) &= z_j^{(\ell-1)}\sigma_\ell'(a_i^{(\ell)})\big[\textstyle\sum_k\delta_k^{(\ell+1)}\sigma_{\ell+1}'(a_k^{(\ell+1)})\big], \forall 1 < \ell < L - 1, \\
\delta_{\text{in}}(w_{ij}^{(L-1)}) &= 0, \\
\delta_{\text{in}}(w_{ij}^{(L)}) &= 0.
\end{aligned}
$$

and the out-degrees

$$
\begin{aligned}
\delta_{\text{out}}(w_{ij}^{(1)}) &= 0, \\
\delta_{\text{out}}(w_{ij}^{(\ell)}) &= \big[\textstyle\sum_m z_m^{(\ell-2)}\big]\sigma_{\ell-1}'(a_j^{(\ell-1)})\delta_i^{(\ell)}\sigma_\ell'(a_i^{(\ell)}), \forall 1 < \ell < L - 1, \\
\delta_{\text{out}}(w_{ij}^{(L-1)}) &= \big[\textstyle\sum_m z_m^{(L-3)}\big]\sigma_{L-2}'(a_j^{(L-2)})\delta_i^{(L-1)}\sigma_{L-1}'(a_i^{(L-1)}), \\
\delta_{\text{out}}(w_{ij}^{(L)}) &= \big[\textstyle\sum_m z_m^{(L-2)}\big]\sigma_{L-1}'(a_j^{(L-1)})(z_i^{(L)} - y_i).
\end{aligned}
$$

It is easy to derive

$$
\begin{aligned}
\boldsymbol{\delta}_{\text{in}}^T\boldsymbol{\delta}_{\text{out}} &= \textstyle\sum_{i,j}\sum_{1<\ell<L-1}\delta_{\text{in}}(w_{ij}^{(\ell)})\delta_{\text{out}}(w_{ij}^{(\ell)}) \\
&= \textstyle\sum_{i,j}\sum_{1<\ell<L-1}\big[\sum_m z_m^{(\ell-2)}\big]z_j^{(\ell-1)}\sigma_{\ell-1}'(a_j^{(\ell-1)})[\sigma_\ell'(a_i^{(\ell)})]^2\delta_i^{(\ell)}\big[\sum_k\delta_k^{(\ell+1)}\sigma_{\ell+1}'(a_k^{(\ell+1)})\big], \\
&= \textstyle\sum_{1<\ell<L-1}[\mathbf{1}^T z^{(\ell-2)}] \times \mathbf{1}^T[z^{(\ell-1)}\odot\boldsymbol{\sigma}_{\ell-1}'] \times \mathbf{1}^T[\boldsymbol{\delta}^{(\ell)}\odot\boldsymbol{\sigma}_\ell'^2] \times \mathbf{1}^T[\boldsymbol{\delta}^{(\ell+1)}\odot\boldsymbol{\sigma}_{\ell+1}'].
\end{aligned}
$$

Now, we move forward to compute the total degree

$$
\begin{aligned}
\mathbf{1}^T\boldsymbol{\delta}_{\text{in}} &= \textstyle\sum_{ij} z_j^{(0)}\sigma_1'(a_i^{(1)})\big[\sum_k\delta_k^{(2)}\sigma_2'(a_k^{(2)})\big] + \sum_{ij}\sum_{1<\ell<L-1} z_j^{(\ell-1)}\sigma_\ell'(a_i^{(\ell)})\big[\sum_k\delta_k^{(\ell+1)}\sigma_{\ell+1}'(a_k^{(\ell+1)})\big], \\
&= [\mathbf{1}^T z^{(0)}] \times [\mathbf{1}^T\boldsymbol{\sigma}_1'] \times \mathbf{1}^T[\boldsymbol{\delta}^{(2)}\odot\boldsymbol{\sigma}_2'] + \textstyle\sum_{1<\ell<L-1}[\mathbf{1}^T z^{(\ell-1)}] \times [\mathbf{1}^T\boldsymbol{\sigma}_\ell'] \times \mathbf{1}^T[\boldsymbol{\delta}^{(\ell+1)}\odot\boldsymbol{\sigma}_{\ell+1}'], \\
&= \textstyle\sum_{1\leq\ell<L-1}[\mathbf{1}^T z^{(\ell-1)}] \times [\mathbf{1}^T\boldsymbol{\sigma}_\ell'] \times \mathbf{1}^T[\boldsymbol{\delta}^{(\ell+1)}\odot\boldsymbol{\sigma}_{\ell+1}'].
\end{aligned}
$$

The definitions of in-degree and out-degree ensure that $\mathbf{1}^T\boldsymbol{\delta}_{\text{in}} = \mathbf{1}^T\boldsymbol{\delta}_{\text{out}}$ must hold. Let's prove it:

$$
\begin{aligned}
\mathbf{1}^T\boldsymbol{\delta}_{\text{out}} &= \textstyle\sum_{i,j}\big[\sum_{1<\ell\leq L-1}\sum_m z_m^{(\ell-2)}\sigma_{\ell-1}'(a_j^{(\ell-1)})\delta_i^{(\ell)}\sigma_\ell'(a_i^{(\ell)}) + \sum_m z_m^{(L-2)}\sigma_{L-1}'(a_j^{(L-1)})(z_i^{(L)} - y_i)\big], \\
&= \textstyle\sum_{1<\ell\leq L-1}[\mathbf{1}^T z^{(\ell-2)}] \times [\mathbf{1}^T\boldsymbol{\sigma}_{\ell-1}'] \times \mathbf{1}^T[\boldsymbol{\delta}^{(\ell)}\odot\boldsymbol{\sigma}_\ell'] = \mathbf{1}^T\boldsymbol{\delta}_{\text{in}}.
\end{aligned}
$$

With the fact that $\boldsymbol{\sigma}_\ell'^2 = \boldsymbol{\sigma}_\ell'$ for ReLU, according to Eq. 3, we have

$$
\beta_{\text{eff}} = \frac{\sum_{\ell=2}^{L-2}[\mathbf{1}^T z^{(\ell-2)}] \times \mathbf{1}^T[z^{(\ell-1)}\odot\boldsymbol{\sigma}_{\ell-1}'] \times \mathbf{1}^T[\boldsymbol{\delta}^{(\ell)}\odot\boldsymbol{\sigma}_\ell'] \times \mathbf{1}^T[\boldsymbol{\delta}^{(\ell+1)}\odot\boldsymbol{\sigma}_{\ell+1}']}{\sum_{\ell=2}^{L-1}[\mathbf{1}^T z^{(\ell-2)}] \times [\mathbf{1}^T\boldsymbol{\sigma}_{\ell-1}'] \times \mathbf{1}^T[\boldsymbol{\delta}^{(\ell)}\odot\boldsymbol{\sigma}_\ell']}.
$$

## C DERIVATION OF ADJACENCY MATRIX $P$ OF $G_B$

The right hand side (RHS) of Eq. 6 is a function of $W^{(\ell+1)}$, and denoted as $F(W^{(\ell+1)})$. Here we derive the strength of the impact from $W^{(\ell+1)}$ and other weights $W^{(-\ell)} = (W^{(0)}, W^{(1)}, \ldots, W^{(\ell)}, W^{(\ell+2)}, \ldots, W^{(L)})$ for building the edge dynamics. Let $W = (W^{(\ell+1)}, W^{(-\ell)})$ and $F(W) = dW^{(\ell)}/dt$. We denote $\hat{W}^{(-\ell)}$ as the current states of $W^{(-\ell)}$), $W^{*(\ell+1)}$ as an equilibrium point, and $W^* = (W^{*(\ell+1)}, \hat{W}^{(-\ell)})$. According to the Taylor expansion, we linearize $F$ at $W^*$ and have

$$dW^{(\ell)}/dt \approx F(W^*) + \frac{\partial F(W^{*(\ell+1)}, \hat{W}^{(-\ell)})}{\partial W^{(\ell+1)}}(W^{(\ell+1)} - W^{*(\ell+1)}) + \frac{\partial F(W^{*(\ell+1)}, \hat{W}^{(-\ell)})}{\partial W^{(-\ell)}}(W^{(-\ell)} - \hat{W}^{(-\ell)}).$$

The last term on the RHS can be cancelled out when the realizations of $W^{(-\ell)}$ take the current states of $W^{(-\ell)}$, i.e. $\hat{W}^{(-\ell)}$. The gradient is simplified as

$$dW^{(\ell)}/dt \approx F(W^*) + \frac{\partial F(W^{*(\ell+1)}, \hat{W}^{(-\ell)})}{\partial W^{(\ell+1)}}(W^{(\ell+1)} - W^{*(\ell+1)}).$$

The term $\partial F(W^{*(\ell+1)}, \hat{W}^{(-\ell)})/\partial W^{(\ell+1)} = \partial^2 \mathcal{C}(W^{*(\ell+1)}, \hat{W}^{(-\ell)})/\partial W^{(\ell)}\partial W^{(\ell+1)}$ effectively captures the interaction strengths between $W^{(\ell)}$ and $W^{(\ell+1)}$, because it measures how much $F$ is affected by a unit perturbation on $W^{(\ell+1)}$. Usually, $W^{*(\ell+1)}$ are not available before updating $W^{(\ell+1)}$ following the update of $W^{(\ell)}$, we use the current states of $W^{(\ell+1)}$ instead. The system can be viewed as a realization of the general Eq. 1, with linear $f(W^{(\ell)}) = F(W^*)$ and $g(W^{(\ell)}, W^{(\ell+1)}) = W^{*(\ell+1)} - W^{(\ell+1)}$. Now, we can immediately have the adjacency matrix $P$ of $G_B$ with $P^{(l,l+1)} = \partial^2\mathcal{C}(W^{(\ell+1)}, \hat{W}^{(-\ell)})/\partial W^{(\ell)}\partial W^{(\ell+1)}, \forall 1 \leq \ell \leq L$.

## D PROOF OF THEOREM 1

The second order gradient $P^{(l,l+1)} = \partial^2\mathcal{C}/\partial W^{(\ell)}\partial W^{(\ell+1)}$ is proposed to measure the interaction strength between $W^{(\ell)}$ and $W^{(\ell+1)}$, $\forall 1 \leq \ell \leq L$. Considering an MLP, and assume that each activation function $\sigma_\ell$ is ReLU for $\ell < L$, when $G_A$ converges, $\nabla_W^{(\ell)}$ vanishes, i.e., $\nabla_W^{(\ell)} = (\delta^{(\ell)} \odot \sigma'^\ell)z^{(\ell-1)T} = \mathbf{0}$ (Eq. 12 in Appendix B). It indicates that $(\delta^{(\ell)} \odot \sigma'^\ell)_i z_j^{(\ell-1)} = 0$, i.e., either $(\delta^{(\ell)} \odot \sigma'^\ell)_i = 0$ or $z_j^{(\ell-1)} = 0, \forall (i,j)$. According to Eq. 10, the numerator involves the product of terms $\delta^{(\ell)} \odot \sigma'^\ell$ and $z^{(\ell-1)}$, which are zeros[7], so $\beta_{\text{eff}} = 0$.

## E BAYESIAN RIDGE REGRESSION

Ridge regression introduces an $\ell_2$-regularization to linear regression, and solves the problem

$$\arg\min_{\theta}(\boldsymbol{y} - X\boldsymbol{\theta})^T(\boldsymbol{y} - X\boldsymbol{\theta}) + \lambda\|\boldsymbol{\theta}\|_2^2, \tag{17}$$

where $X \in \mathcal{R}^{n\times d}, \boldsymbol{y} \in \mathcal{R}^n, \boldsymbol{\theta} \in \mathcal{R}^d$ is the associated set of coefficients, the hyper-parameter $\lambda > 0$ controls the impact of the penalty term $\|\boldsymbol{\theta}\|_2^2$.

**Bayesian ridge regression** introduces uninformative priors over the hyper-parameters of the model, and estimates a probabilistic model of the problem in Eq. 17. Usually, the ordinary least squares method posits the conditional distribution of $\boldsymbol{y}$ to be a Gaussian, i.e., $p(\boldsymbol{y}|X, \boldsymbol{\theta}) = \mathcal{N}(\boldsymbol{y}|X\boldsymbol{\theta}, \sigma^2 I_d)$, where $\sigma > 0$ is a hyper-parameter to be tuned, and $I_d$ is a $d\times d$ identity matrix. Moreover, if we assume a spherical Gaussian prior $\boldsymbol{\theta}$, i.e., $p(\boldsymbol{\theta}) = \mathcal{N}(\boldsymbol{\theta}|0, \tau^2 I_d)$, where $\tau > 0$ is another hyper-parameter to be estimated from the data at hand. According to Bayes' theorem, $p(\boldsymbol{\theta}|X, \boldsymbol{y}) \propto p(\boldsymbol{\theta})p(\boldsymbol{y}|X, \boldsymbol{\theta})$, the estimates of the model are made by maximizing the posterior distribution $p(\boldsymbol{\theta}|X, \boldsymbol{y})$, i.e.,

$$\arg\max_{\theta} \log p(\boldsymbol{\theta}|X, \boldsymbol{y}) = \arg\max_{\theta} \log\mathcal{N}(\boldsymbol{y}|X\boldsymbol{\theta}, \sigma^2 I_d) + \log\mathcal{N}(\boldsymbol{\theta}|\mathbf{0}, \tau^2 I_d),$$

which is a maximum-a-posteriori (MAP) estimation of the ridge regression when $\lambda = \sigma^2/\tau^2$. All $\boldsymbol{\theta}$, $\lambda$ and $\tau$ are estimated jointly during the fit of the model, and $\sigma = \tau\sqrt{\lambda}$.

---

[7]A small constant $\varepsilon$ is added to the denominator of $\beta_{\text{eff}}$ to avoid division by zero.

To estimate $I = h(\beta_{\text{eff}}; \boldsymbol{\theta})$, we use scikit-learn[8], which is built on the algorithm described in Tipping (2001) updating the regularization parameters $\lambda$ and $\tau$ according to MacKay (1992).

## F  RANKING PERFORMANCE ON ALL FIVE DATASETS

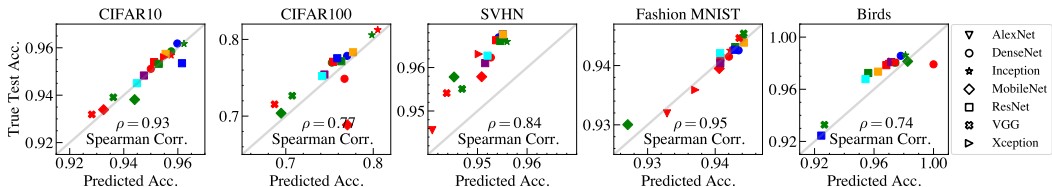

Figure F.4: Predictions of the validation accuracy of pre-trained models on all five datasets based on $\beta_{\text{eff}}$ v.s. true test accuracy of these models after fine-tuning for $T = 50$ epochs. The Spearman's ranking correlation $\rho$ is used to quantify the performance in model selection. Each shape is associated with one type of pre-trained models. Distinct models of the same type are marked in different colors. To be noted, each includes AlexNet in computing $\rho$s.

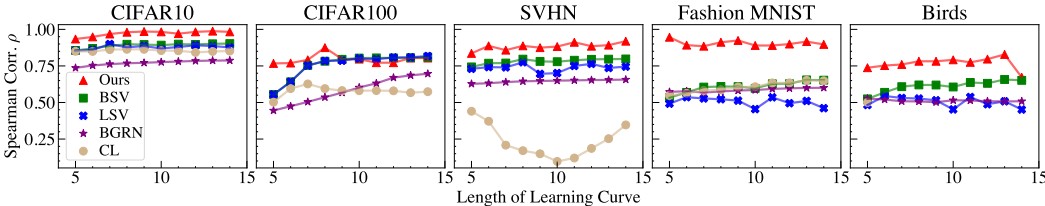

Figure F.5: A comparison between our $\beta_{\text{eff}}$ based approach and the baselines in model ranking.

## G  RUNNING TIME ANALYSIS

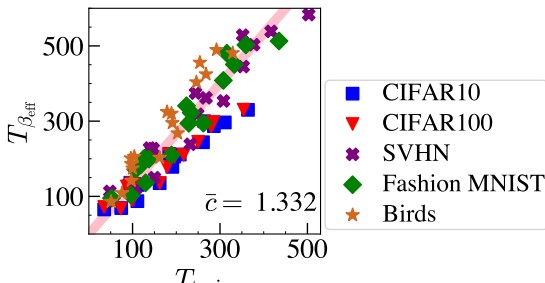

Figure G.6: Training time per epoch versus computing time for $\beta_{\text{eff}}$ per epoch over all 17 pre-trained models and five datasets discussed in the main text. Each data point is associated with one pre-trained model over one dataset. The relative cost of our approach in computing $\beta_{\text{eff}}$ with respect to training more epochs can be measured by $c = T_{\beta_{\text{eff}}}/T_{\text{train}}$. On average, it is $\bar{c} \approx 1.3$ (slope of the pink line).

Table G.2: Running time (in seconds) in learning curve prediction, including the predictor estimation (if necessary, e.g., Ours, CL and BGRN).

| Dataset | Ours | BGRN | LSV | BSV | CL |
|---|---|---|---|---|---|
| CIFAR10 | 0.491 | 0.610 | 0.059 | 0.049 | 3966.128 |
| CIFAR100 | 0.414 | 0.628 | 0.051 | 0.045 | 5256.478 |
| SVHN | 0.506 | 0.607 | 0.074 | 0.044 | 4690.507 |
| Fashion MNIST | 0.493 | 0.625 | 0.057 | 0.046 | 4552.194 |
| Birds | 0.460 | 0.636 | 0.071 | 0.044 | 4734.992 |

---

[8]https://scikit-learn.org/stable/modules/generated/sklearn.linear_model.BayesianRidge.html

## H    MEAN-FIELD APPROACH

We summarize the main idea of the mean-field approach developed by Gao et al. (2016), and show how Eq. 3 is obtained (Jiang et al., 2020a;b).

Based on the notations described in Section 3, we consider a vertex $i$ and the interaction term $\sum_j P_{ij} g(x_i, x_j)$ in Eq. 1, where $P_{ij}$ is the influence $j$ has on $i$. Similarly, $i$ influences $j$ with a weight $P_{ji}$. We define the in-degree $\delta_i^{\text{in}} = \sum_j P_{ij}$ and the out-degree $\delta_i^{\text{out}} = \sum_j P_{ji}$. The interaction term can be rewritten as

$$\sum_j P_{ij} g(x_i, x_j) = \delta_i^{\text{in}} \frac{\sum_j P_{ij} g(x_i, x_j)}{\sum_k P_{ik}}. \tag{18}$$

Here the in-degrees $\boldsymbol{\delta}^{\text{in}}$ captures the idiosyncratic part, and the average $g(\cdot, \cdot)$ captures the network effect. The mean-field approximation is to replace local averaging with global averaging, which approximates the network impact on a vertex as nearly homogeneous. Specifically, we can get

$$\frac{\sum_j P_{ij} g(x_i, x_j)}{\sum_k P_{ik}} \approx \frac{\sum_{ij} P_{ij} g(x_i, x_j)}{\sum_{ik} P_{ik}} = \frac{\mathbf{1}^T P g(x_i, \boldsymbol{x})}{\mathbf{1}^T P \mathbf{1}}, \tag{19}$$

where the vector $g(x_i, \boldsymbol{x})$ has the $j$th component $g(x_i, x_j)$. A linear operator

$$\mathcal{L}_P(\boldsymbol{z}) = \frac{\mathbf{1}^T P}{\mathbf{1}^T P \mathbf{1}} \boldsymbol{z} = \frac{\boldsymbol{\delta}^{\text{out}} \cdot \boldsymbol{z}}{\boldsymbol{\delta}^{\text{out}} \cdot \mathbf{1}} \tag{20}$$

is defined for a weighted average of the entries in $\boldsymbol{z}$. The mean-field approximation gives

$$\dot{x}_i = f(x_i) + \delta_i^{\text{in}} \mathcal{L}_P[g(x_i, \boldsymbol{x})]. \tag{21}$$

In the first order linear approximation, we can take the $\mathcal{L}_P$-average inside $g$. The average of external interactions is approximately the interaction with its average, i.e. $\mathcal{L}_P[g(x_i, \boldsymbol{x})] \approx g(x_i, \mathcal{L}_P(\boldsymbol{x}))$ and

$$\dot{x}_i = f(x_i) + \delta_i^{\text{in}} g(x_i, \mathcal{L}_P(\boldsymbol{x})), \tag{22}$$

where $\mathcal{L}_P(\boldsymbol{x})$ is a global state. Let $x_{\text{av}} \triangleq \mathcal{L}_P(\boldsymbol{x})$. Applying $\mathcal{L}_P$ to both sides of Eq. 22 gives

$$\dot{x}_{\text{av}} = \mathcal{L}_P[f(\boldsymbol{x})] + \mathcal{L}_P[\boldsymbol{\delta}^{\text{in}} g(\boldsymbol{x}, x_{\text{av}})]. \tag{23}$$

According to the extensive discussion and tests in (Gao et al., 2016), the in-degrees $\boldsymbol{\delta}^{\text{in}}$ and the interaction with the external $x_{\text{av}}$ are roughly uncorrelated, so the $\mathcal{L}_P$-average of the product is roughly the product of $\mathcal{L}_P$-averages. Therefore, $\mathcal{L}_P[\boldsymbol{\delta}^{\text{in}} g(\boldsymbol{x}, x_{\text{av}})] \approx \mathcal{L}_P(\boldsymbol{\delta}^{\text{in}}) \mathcal{L}_P[g(\boldsymbol{x}, x_{\text{av}})]$. Using the first order linear approximation, we take the $\mathcal{L}_P$-average inside $f$ and $g$

$$\dot{x}_{\text{av}} = f(\mathcal{L}_P(\boldsymbol{x})) + \mathcal{L}_P(\boldsymbol{\delta}^{\text{in}}) g(\mathcal{L}_P(\boldsymbol{x}), x_{\text{av}}). \tag{24}$$

Therefore, we have

$$\dot{x}_{\text{av}} = f(x_{\text{av}}) + \beta_{\text{eff}} g(x_{\text{av}}, x_{\text{av}}),$$

where $\beta_{\text{eff}} = \mathcal{L}_P(\boldsymbol{\delta}^{\text{in}})$ is the resilience metric, and its steady-state is the effective network impact $x_{\text{eff}}$, satisfying $\dot{x}_{\text{eff}} = f(x_{\text{eff}}) + \beta_{\text{eff}} g(x_{\text{eff}}, x_{\text{eff}}) = 0$.

## I    CORE PROCEDURE

Our framework is built on several different techniques and the related contents are dispersed in different sections. Here we briefly summarize the core idea of this paper and show how these sections are organized, see Fig. I.7 for a flowchart of our core procedure.

We view the NN training as a dynamical system, and directly model the evolving of the trainable weights in the SGD based training as a set of differential equations (Section 3), characterized by a general dynamics in Eq. 1. Usually, it is convenient to study the dynamics of agents (trainable weights in our case) on a regular network, where each node represents an agent in the dynamical system and the interactions of agents are governed by Eq. 1. Many powerful techniques have been developed in network science and dynamical systems, e.g. the universal metric $\beta_{\text{eff}}$ developed by Gao et al. (2016) to quantify and categorize various types of networks, including biological neural networks

(Shu et al., 2021). Because of the generality of the metric, we analyze how it looks on artificial neural networks which are designed to mimic the biological counterpart for general intelligence. Therefore, an analogue system of the trainable weights under the context of the general dynamics is set up in our framework. To the end, we build a line graph for the trainable weights (Fig. 1a and Section 4.1) and "rewrite" (Section 4.2 and Appendix C) the training dynamics in the form of Eq. 1, which includes a self-driving force $f(\cdot)$, an external driving force $g(\cdot, \cdot)$ and an adjacency matrix $P$ (Eqs. 8 & 9).

The reformulated training dynamics yields a simple yet powerful property. It is proved that as the neural network converges, $\beta_{\text{eff}}$ approaches zero (Theorem 1 in Section 4.3, also one of our primary contributions). As shown in Fig. 1(c) and Section 4.3, we exploit the property to predict the final accuracy of a neural network model with a few observations during the early phase of the training, and apply it to select the pre-trained models (Algorithm 1 in Section 4.4). Generally speaking, the metric $\beta_{\text{eff}}$ should be calculated for the entire neural network. However, because many state-of-the-art neural network models have large-scale trainable weights. If all layers are considered, it will be prohibitive to compute the associated $\beta_{\text{eff}}$. We make a compromise, and estimate $\beta_{\text{eff}}$ of the entire network using $\beta_{\text{eff}}$ of the NCP unit (i.e., a partial part of the entire network, see the second to the last sentence of Section 4.3). It's confirmed from our empirical experiments (Section 5) that the simplified, lightweight version of $\beta_{\text{eff}}$ is still effective in predicting the final accuracy of the entire network.

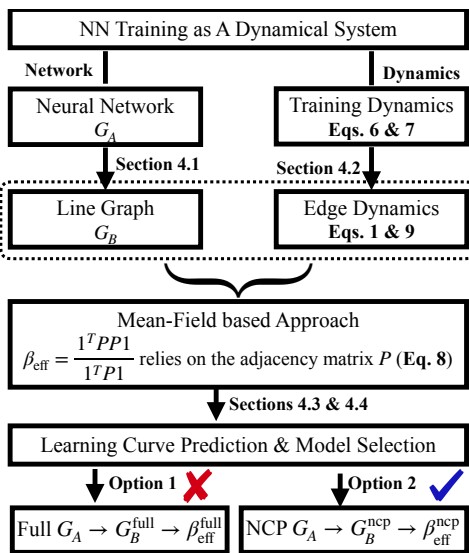

Figure I.7: A flowchart of the core procedure of our framework.

The metric $\beta_{\text{eff}}$ developed by Gao et al. (2016) is universal to characterize different types of networks. Although our framework utilizes the metric, our application to artificial neural network training dynamics and the related theoretical results as specified by Theorem 1 are novel. Specifically, it is applied to study the NN training (Section 3) and predict the final accuracy of an NN with a few observations during the early phase of the training (Fig. 1c). But $\beta_{\text{eff}}$ relies on the adjacency matrix $P$ of $G_A$ (Eq. 3). To derive $P$, we resort to a reformulation (Section 4.2 and Appendix C) of the training dynamics in the same form of the general dynamics (Eq. 1). One issue in calculating $\beta_{\text{eff}}$ is the complexity if the entire $G_A$ is considered. As a resolution, we propose to use the lightweight $\beta_{\text{eff}}$ of the NCP unit – a partial of $G_A$ – to predict the performance of the entire network (Sections 4.3 & 4.4).

