# OpenReview forum: "Neural Capacitance: A New Perspective of Neural Network Selection via Edge Dynamics"
_ICLR.cc/2022/Conference — ICLR 2022 Submitted_

### Official Review · Reviewer_EL9b · 2021-11-02

**Correctness:** 4
**Technical Novelty And Significance:** 3
**Empirical Novelty And Significance:** 3
**Recommendation:** 6
**Confidence:** 2

**Main Review:**

I am not an expert on this research topic. maybe I cannot give a precise review.

Strengths:

The whole paper is written in high-quality and a clear manner.

To validate the outstanding performance of the proposed method, the authors conduct experiments on many popular CNN models.

Weakness:

I am not 100% sure about the performance, because the author reports the accuracy with figures rather than tables.

The authors conduct the experiments on well-known models, what about applying this framework to select the subnetwork on many NAS benchmarks e.g., [1].


[1]. Hw-nas-bench: Hardware-aware neural architecture search benchmark


**Summary Of The Paper:**

This paper proposes a framework to select the neural networks for downstream tasks. To identify the better generalization model, the authors propose a new metric (Neural Capacitance, NCP) to predict precise learning curves. And the authors provide the theoretical explanations for NCP.

Then the authors have verified the advantages of the proposed method when being applied to different datasets (CIFAR10, CIFAR100, SVHN, Fashion MNIST, Birds) with 17 CNN models.

**Summary Of The Review:**

NA

---

> ### Author Response · Authors · 2021-11-21
> **Responses to Reviewer EL9b's Comments**
>
> > Weakness: I am not 100% sure about the performance, because the author reports the accuracy with figures rather than tables.
>
> Please see our general responses to comments concerning (II) **readability of figures on performance**.
>
> > The authors conduct the experiments on well-known models, what about applying this framework to select the subnetwork on many NAS benchmarks e.g., [1] Hw-nas-bench: Hardware-aware neural architecture search benchmark.
>
> We appreciate the reviewer for bringing HW-NAS-Bench [1] to us, and the valuable suggestion of applying our framework to NAS benchmarks to select the best subnetwork. The direction is not within the scope of this paper since we focus on model selection in transfer learning settings. We are currently exploring this direction and planning to extend the framework to general NAS benchmarks or hardware-aware NAS research, even general neural networks that are trained from scratch. However, all these scenarios may affect our reformulation of the training dynamics (Section 4.2) and it will be necessary to make some non-trivial adjustments, such as a new definition of the adjacency matrix $P$ (Eq. 8) or another way to map the neural networks to the regular network for the trainable weights. Accordingly, we **cite the reference and add such extension as one research direction to the conclusion section** in the revised version.

---

### Official Review · Reviewer_acZh · 2021-11-04

**Correctness:** 2
**Technical Novelty And Significance:** 3
**Empirical Novelty And Significance:** Not applicable
**Recommendation:** 5
**Confidence:** 3

**Main Review:**

**Strengths:**
1. Application of dynamic systems theory to NNs.

While the theory described in the paper is not novel, the application of it to the task of predicting model performance is novel as far as I can tell.

**Weaknesses:**

1. Clarity

The paper uses multiple prior ideas from dynamical systems and machine learning literature but many of these choices are not well motivated or explained. This makes it difficult for a broad ML audience to understand the paper and its significance.
- The paper draws heavily from theory described in Gao et al 2016 which proposes a method to convert the reslience function on a multi-dimensional system to a single dimensional function x_eff where the critical points B_eff are a one-dimensional value. Gao et al 2016 describes this approach for a non-neural network system and it is unclear why the assumptions introduced in this paper apply to neural networks.

The simplifying assumption used in the paper is that the dynamics of a node in a complex multi-dimensional system can be characterized by the average nearest-neighbor activity. However, in a neural network, perturbing a single weight doesn't only affect the gradients of the immediate neighbors of the node. Thus, it is unclear why this approximation is appropriate for neural networks. This is briefly touched upon in section 4.2 where it is stated that "the others’ contribution as a whole is implicitly encoded in the activation gradient" however the connection to the original theory described ini Gao et al 2016 and why this is an appropriate simplification needs more explanation.

- Next, the motivation for the NCP is not discussed in the paper. By initializing it randomly and freezing it, a random amount of fixed noise is introduced into the process and it is unclear what the purpose of this is. Additionally, is this randomly initialized each time for all the experiments? And if so, are the same experiments repeated with different initializations to control for randomness?

- What is the motivation of using the second order gradient of C to quantify the interaction strength?

2. Correctness
- In Appendix B, section D "PROOF OF THEOREM 1", it is claimed that the gradient vanishes when one of the terms of the product is 0. Then, it is stated that when either goes to 0, the numerator in beta_eff goes to 0. However, it appears that if the gradient (left) term goes to 0, the denominator also goes to 0 since the term is also present in the denominator product. This will result in beta_eff being undefined. Please clarify whether this is accurate.

3. Details
- In the last sentence of section 4.2, the self-dynamics part, f(w_i) is defined as F(w_i*), however, I couldn't find where this was defined. What does this correspond to in the context of neural networks? Additionally, why is g(w_i, w_j) = w_j - w_j*?

4. Figures
- The figures are small and difficult to read. A better presentation of the results such as a table could be added to the main paper while the figures are either added to the appendix or enlarged.

**Summary Of The Paper:**

The paper formulates neural network training as a dynamical system and uses existing theory to formulate a metric (beta_eff) to predict how a pre-trained model will perform on a downstream task without requiring to train the model to convergence.

**Summary Of The Review:**

The paper introduced a novel formulation of neural networks based on prior dynamical systems theory for an important machine learning problem. However, there are some parts of the paper that require further explanation and in its current state, I think it is fairly difficult for a broad ML audience to understand the paper and it's significance.

---

> ### Author Response · Authors · 2021-11-20
> **Responses (I) to Reviewer acZh's Comments**
>
> > Weaknesses: Clarity. The paper uses multiple prior ideas from dynamical systems and machine learning literature but many of these choices are not well motivated or explained. This makes it difficult for a broad ML audience to understand the paper and its significance.
>
> We regret that the reviewer found that the paper is difficult to understand due to the clarity issue. We made some major changes in the revised version. Please see our general responses to comments concerning (I) **overall structure and motivations**.
>
> > Gao et al 2016 describes this approach for a non-neural network system and it is unclear why the assumptions introduced in this paper apply to neural networks. The simplifying assumption used in the paper is that the dynamics of a node in a complex multi-dimensional system can be characterized by the average nearest-neighbor activity. However, in a neural network, perturbing a single weight doesn't only affect the gradients of the immediate neighbors of the node. Thus, it is unclear why this approximation is appropriate for neural networks.
>
> We appreciate the reviewer for an insightful summary of the mean-field based approach developed in [Gao et al. 2016]. The approach works for general networks, including biological neural networks [Shu et al. 2021]. The artificial neural networks are designed to simulate the biological ones for general intelligence. They are not an exception, and we can still apply the approach developed by Gao et al. (2016).
>
> We agree with the reviewer that the impacts of perturbing a single weight in a neural network will affect the gradients of many other weights besides those of the immediate neighbors, but this approximation is appropriate because of the following reasons.
>
> (1) The impacts of the perturbed links may vary wildly across layers and types of neural networks. Prior works on deep compression or weight pruning [Hinton et al. 2015, Han et al. 2015, Molchanov et al. 2017, Frankle & Carbin 2019] provide numerous evidence that the removal of a moderate percentage of edges from the neural networks (especially overparameterized ones) does not negatively affect the overall performance too much [Nowak & Lelowicz 2021].
>
> (2) In a networked dynamical system studied in [Gao et al. 2016], the state of a node not only depends on the states of its nearest neighbors but also other nodes that can reach it, which is similar to what the reviewer described about neural networks. In a network environment, a perturbation does not stay localized, but can reach other nodes, following the distance-dependent correlation function [Barzel et al. 2013, Duan et al. 2019]. We still can use this mean-field approximation to estimate its dynamics.
>
>  (3) Our approximation is appropriate if we can model the neural network training as a networked dynamical system. From the **network aspect** of the trainable weights, our mapping scheme from the neural network $G_A$ to the associated line graph $G_B$ intactly preserves the hierarchical topological relation of $G_A$ via trainable weights in terms of a regular network. In $G_B$, no direct edge between two nodes (i.e., trainable weights) does not imply that there is no impact from one to another. Also, it will be prohibitive to directly study a (nearly) complete graph over all trainable weights, regardless of the types of the impacts, direct or indirect. As for **the governing dynamics** over $G_B$, we reformulate it from the training dynamics by emphasizing the direct impacts of nodes (i.e. trainable weights) in $G_B$, a common strategy adopted in studying a variety of dynamical systems in network science. Specifically, we apply Taylor expansion to “rewrite” the training dynamics as Eq. 9, including an explicit expression of the internal $f(\cdot)$, the external $g(\cdot,\cdot)$ driving dynamics and the adjacency matrix $P$. The reformulation precisely fits the general dynamics in Eq. 1. Since the training dynamics can be written into the general form, the framework developed by Gao et al. (2016) can be safely applied to artificial neural networks.
>
> (4) Due to the high demand in memory and computation, a full analysis of all pairwise interactions is challenging. An approximation is usually required in many applications. For example, the popular 2nd-order optimization approach K-FAC [Martens & Grosse 2015, Ba et al 2017, Bernacchia et al 2019] over deep neural networks assumes that 1) weight-gradients in different layers are uncorrelated, and 2) the layer input and output-gradients are independent. These approximations sometimes may not hold, but they are necessary simplifications to deal with complicated deep neural networks.

---

> > ### Author Response · Authors · 2021-11-20
> > **Responses (II) to Reviewer acZh's Comments**
> >
> > > The motivation for the NCP is not discussed in the paper.
> >
> > We discussed our motivation in Section 4.3 (see **the paragraph below Theorem 1**). Here we rephrase it as “For an MLP $G_A$, it is possible to derive an analytical form of $\beta_{\rm eff}$. However, it becomes extremely complicated for a deep NN with multiple convolutional layers. [...] Considering the number of parameters, it is still computationally expensive, and prohibitive to calculate a $\beta_{\rm eff}$ for the entire $G_A$. Because of this, we seek to derive a surrogate from a partial of $G_A$. As shown in Section 4.4, we insert a neural capacitance probe (NCP) unit, i.e., putting additional layers on top of the beheaded $G_A$ (excluding the original output layer), and estimate the predictive ability of the entire $G_A$ using $\beta_{\rm eff}$ of the NCP unit.”  Also, we provide a detailed explanation of our motivation in the general responses, see (I) overall structure and motivations.
> >
> > > By initializing it randomly and freezing it, a random amount of fixed noise is introduced into the process and it is unclear what the purpose of this is. Additionally, is this randomly initialized each time for all the experiments? And if so, are the same experiments repeated with different initializations to control for randomness?
> >
> > (1) Purpose. Please see our general responses to comments concerning (II) **purpose of random initialization**.
> >
> > (2) Repeated random initialization. Yes. The models are randomly initialized each time for all the experiments. Also, we repeat the experiments with different initializations on each pre-trained model over each dataset for 20 times (see the **Experimental setup in Section 5**).
> >
> > > What is the motivation of using the second order gradient of C to quantify the interaction strength?
> >
> > Considering a better readability,  in our original manuscript, we moved the detailed mathematical derivation into **Appendix C**. The derivation is motivated to reformulate the training dynamics to the same form of the general dynamics as described in Eq. 1 for applying $\beta_{\rm eff}$ to analyze the behavior of NN training. The general dynamics is characterized by three components: a self-driving force $f(\cdot)$, an external driving force $g(\cdot,\cdot)$ and an adjacency matrix $P$ (**Eqs. 8 & 9 in Section 4.2**). To get the adjacency matrix $P$, we decouple the nonlinear gradient by applying the linearization at $W^*$. The 2nd order gradients of $\mathcal C$ (Eq. 8), in essence,  are natural products of linearization, and each entry of $P$ is associated with the interaction strength between a pair of nodes (i.e., trainable weights of $G_A$). We would like to emphasize that this idea is novel, and we intend to explore it further in the future.
> >
> > > When the gradient vanishes, a possible 0/0 in $\beta_{\rm eff}$ may happen. Please clarify whether this is accurate.
> >
> > We appreciate the reviewer for careful reading and identifying this issue. In our implementation, we added a small value $\varepsilon$ to the denominator to avoid a possible 0/0. Moreover, because of the numerical precision, it is rare to reach 0/0. Accordingly, we added a footnote to specify how to handle the rare case in the revised manuscript (see **Appendix D**).

---

> > > ### Author Response · Authors · 2021-11-21
> > > **Responses (III) to Reviewer acZh's Comments**
> > >
> > > > Details: In the last sentence of section 4.2, the self-dynamics part, f(w_i) is defined as F(w_i*), however, I couldn't find where this was defined. What does this correspond to in the context of neural networks? Additionally, why is g(w_i, w_j) = w_j - w_j*?
> > >
> > > We appreciate the reviewer for pointing this out. The notations were accidentally omitted from the main paper when we moved some mathematical derivations to Appendix C.
> > >
> > > We view the NN training as a dynamical system (Section 3). Given a neural network $G_A$, we map it to the line graph $G_B$ of trainable weights (Section 4.1), and reformulate the training dynamics in the same form as the general dynamics in Eq. 1, which is characterized by three components, (i) the adjacency matrix $P$, (ii) the self-driving dynamics $f(w_i)$ and (iii) the external driving force $g(x_i, x_j)$.  To analyze the training dynamics with the mean-field based approach, we need to derive these three components. Below we summarize the main idea of the derivation, please refer to Appendix C for more details.
> > >
> > > Let $F(W^{(\ell+1)})$ denote the right hand side of Eq. 6 (i.e. the training dynamics). It is a function of $W^{(\ell+1)}$. Here we derive the strength of the impact from $W^{(\ell+1)}$ and other weights $W^{(-\ell)}=(W^{(0)},W^{(1)},\ldots,W^{(\ell)},W^{(\ell+2)},\ldots,W^{(L)})$ for the edge dynamics (Section 4.2).
> > > Let $W=(W^{(\ell+1)},W^{(-\ell)})$ and $F(W) = dW^{(\ell)}/dt$. We denote $\hat W^{(-\ell)}$ as the current states of $W^{(-\ell)})$, $W^{\*(\ell+1)}$ as an equilibrium point,
> > > and $W^\*=(W^{\*(\ell+1)},\hat W^{(-\ell)})$.  According to the Taylor expansion,
> > > we linearize $F$ at $W^\*$ and have
> > > $$
> > > d W^{(\ell)}/ dt \approx
> > > F(W^\*) + \frac{\partial F(W^{\*(\ell+1)},\hat W^{(-\ell)})}{\partial W^{(\ell+1)}} (W^{(\ell+1)}-W^{\*(\ell+1)}).
> > > $$
> > > The term
> > > $\partial F(W^{\*(\ell+1)},\hat W^{(-\ell)})/\partial W^{(\ell+1)}=\partial^2 \mathcal C(W^{\*(\ell+1)},\hat W^{(-\ell)})/\partial W^{(\ell)}\partial W^{(\ell+1)}$ effectively captures the interaction strengths between $W^{(\ell)}$ and $W^{(\ell+1)}$, because it measures how much $F$ is affected by a unit perturbation on $W^{(\ell+1)}$.
> > >
> > > Usually, $W^{\*(\ell+1)}$ are not available before updating $W^{(\ell+1)}$ following the update of $W^{(\ell)}$, we use the current states of $W^{(\ell+1)}$ instead. The system can be viewed as a realization of the general Eq. 1, with linear $f(W^{(\ell)})=F(W^*)$ and $g(W^{(\ell)},W^{(\ell+1)})=W^{\*(\ell+1)}-W^{(\ell+1)}$.  We concatenate all the weight matrices $W^{(\ell)}$, flatten them into a big vector and discarded their superscripts. For a trainable weight $w_i$, we have
> > > $$
> > > \dot w_i = w_i^\* + \sum_j P_{ij} (w_i - w_j^\*),
> > > $$
> > > where $j$ is any neighbor of $i$, and $P_{ij}=\partial^2 \mathcal C/\partial w_i\partial w_j$. The differential equation is a realization of the general dynamics in Eq. 1, so we can get the associated $f(w_i)=w_i^*$ and $g(w_i,w_j)=w_i-w_j^*$.
> > > To make it relatively easy for the reviewer and other readers to understand the relationship between a specific dynamical system and the general one in Eq. 1, we show two example dynamical systems [Gao et al. 2016].
> > > One example is mutualistic network
> > > $$
> > > \dot x_i = B+x_i(1-x_i/K)(x_i/C-1)+\sum_{j\in V}P_{ij} x_ix_j/(D+E x_i+H x_j).
> > > $$
> > > Rewritten it in the general form, we have $f(x_i)=B+x_i(1-x_i/K)(x_i/C-1)$ and $g(x_i,x_j)=x_ix_j/(D+E x_i+H x_j)$, which is a coupled and non-linear function of a pair of states and very common for many real complex systems.
> > >
> > > Another example is gene regulatory network
> > > $$
> > > \dot x_i = -Bx_i^\alpha + \sum_{j\in V} P_{ij} x_j^h/(x_j^h+1).
> > > $$
> > > We can get the corresponding $f(x_i)=-Bx_i^\alpha$ and $g(x_i,x_j)=x_j^h/(x_j^h+1)$, which is completely independent of $x_i$. Here the external driving dynamics is similar to our case, $g(w_i, w_j) = w_j - w_j*$ is independent of $w_i$ and relies on $w_j$ only.
> > >
> > > > Figure: The figures are small and difficult to read. A better presentation of the results such as a table could be added to the main paper while the figures are either added to the appendix or enlarged.
> > >
> > > Please see our general response to comments concerning (III) **readability of figures on performance**.

---

> > > > ### Author Response · Authors · 2021-11-21
> > > > **References to the Responses to Reviewer acZh's Comments**
> > > >
> > > > **References**:
> > > >
> > > > [Ba et al 2017] Ba, Jimmy, Roger Grosse, and James Martens. "Distributed second-order optimization using Kronecker-factored approximations." ICLR, 2017.
> > > >
> > > > [Bernacchia et al 2019] Bernacchia, Alberto, Máté Lengyel, and Guillaume Hennequin. "Exact natural gradient in deep linear networks and application to the nonlinear case." NIPS, 2019.
> > > >
> > > > [Martens & Grosse 2015] Martens, James, and Roger Grosse. "Optimizing neural networks with kronecker-factored approximate curvature." In International conference on machine learning, pp. 2408-2417. PMLR, 2015.
> > > >
> > > > [Gao et al. 2016] Gao, Jianxi, Baruch Barzel, and Albert-László Barabási. "Universal resilience patterns in complex networks." Nature, 2016.
> > > >
> > > > [Hinton et al. 2015] Hinton, Geoffrey, Oriol Vinyals, and Jeff Dean. "Distilling the knowledge in a neural network." arXiv preprint arXiv:1503.02531, 2015.
> > > >
> > > > [Han et al. 2015] Han, Song, Huizi Mao, and William J. Dally. "Deep compression: Compressing deep neural networks with pruning, trained quantization and huffman coding." ICLR, 2016.
> > > >
> > > > [Molchanov et al. 2017] Molchanov, Pavlo, Stephen Tyree, Tero Karras, Timo Aila, and Jan Kautz. "Pruning convolutional neural networks for resource efficient inference." ICLR, 2017.
> > > >
> > > > [Frankle & Carbin 2019] Frankle, Jonathan, and Michael Carbin. "The lottery ticket hypothesis: Finding sparse, trainable neural networks.", ICLR, 2019.
> > > >
> > > > [Nowak & Lelowicz 2021] Nowak, Mariusz Karol, and Kamil Lelowicz. "Weight Perturbation as a Method for Improving Performance of Deep Neural Networks." In 2021 25th International Conference on Methods and Models in Automation and Robotics (MMAR), pp. 127-132. IEEE, 2021.
> > > >
> > > > [Shu et al. 2021] Shu, Pin, Hong Zhu, Wen Jin, Jie Zhou, Shanbao Tong, and Junfeng Sun. "The Resilience and Vulnerability of Human Brain Networks Across the Lifespan." IEEE Transactions on Neural Systems and Rehabilitation Engineering 29 (2021): 1756-1765.
> > > >
> > > > [Barzel et al. 2013] Barzel, Baruch, and Albert-László Barabási. "Universality in network dynamics." Nature Physics 9.10 (2013): 673-681.
> > > >
> > > > [Duan et al. 2019] Duan, Dongli, et al. "Universal behavior of cascading failures in interdependent networks." Proceedings of the National Academy of Sciences 116.45 (2019): 22452-22457.

---

### Official Review · Reviewer_A1LA · 2021-11-08

**Correctness:** 4
**Technical Novelty And Significance:** 3
**Empirical Novelty And Significance:** 2
**Recommendation:** 5
**Confidence:** 2

**Main Review:**

The proposed method is novel. Experiments show that the proposed neural capacitance more accurately predicts the accuracy of the model. Ablation study have been performed to study the effect of parameters such as starting epochs, size of training set, etc.

Overall, I think the paper is potentially a good paper, though I'm not an expert in this topic and does not fully understand the mathematical derivation in Section 4.2 and 4.3. My major concern is

- I'm not sure if I fully understand NCP units. Why the weights are randomly initialized and freezed during finetuning? And in Figure 1(b) does the output layer of right stack do the same thing as output layer of left stack for classfication tasks? I do not find the relationship between the new layers and what is described in section 4.2 and 4.3. Why two dense layers are chosen to be the architecture of NCP units? I think this part is missing from the paper and should be elaborated to build the connection between the previous section.


**Summary Of The Paper:**

Accurately predicting the performance at early stage is important for efficient model selection without incurring too much computation. The paper proposes a neural capacitance metric as a predictive measure to capture the performance of a model on the downstream task using only a handful of early training results. The metric is derived from a line graph mapped from a neural network by modeling the dynamical system of the network.

**Summary Of The Review:**

Overall, I think the paper proposes an interesting idea to model the training NN as a dynamical system and is potentially a good paper, but the some part of the method needs more elaboration.

---

> ### Author Response · Authors · 2021-11-20
> **Responses to Reviewer A1LA's Comments**
>
> We thank the reviewer for specifying the parts that require more elaboration, and the comments are very helpful in improving our manuscript.
>
> > I'm not sure if I fully understand NCP units. Why are the weights randomly initialized and frozen during fine-tuning?
>
> Please see our general responses to comments concerning (II) **purpose of random initialization**.
>
> > In Figure 1(b) does the output layer of the right stack do the same thing as the output layer of the left stack for classification tasks?
>
> Yes, the output layer of the right stack is added also for classification tasks. But their input and output dimensions may vary depending on the source model architecture and the number of classes in the target domain. For example, the ImageNet classification (source domain) includes 100 classes, but CIFAR10  (target domain) only has 10 classes, so we need an output layer of size 10 for the classification tasks on CIFAR10.
>
> > I do not find the relationship between the new layers and what is described in section 4.2 and 4.3.
>
> We regret that the reviewer did not capture the relationship. We have made some major changes in the revised version. For a detailed explanation of how they are related, please see our general responses to comments concerning (I) **overall structure and motivations**.
>
> > Why are two dense layers chosen to be the architecture of NCP units? I think this part is missing from the paper and should be elaborated to build the connection between the previous section.
>
> We appreciate the reviewer for raising this question. One of our primary goals is to construct a minimal NCP for a well-defined $\beta_{\rm eff}$. According to the derivation in Appendix B, we found that the minimum number of hidden layers required for a well-defined $\beta_{\rm eff}$ is three. According to the common practice in transfer learning, a batch normalization added between two dense layers or convolutional layers is usually beneficial to the performance. We follow the practice, choose two dense layers and two batch normalization layers to be the architecture of the NCP unit. Besides a well-defined $\beta_{\rm eff}$, the overall performance of the pre-trained model on the target domain can be preserved as well.
>
> To avoid any confusion, we add two more sentences in the **Experimental setup of Section 5** to explain the reason for our choice of such architecture, and refer the reader to Appendix C for the related derivation.

---

### Author Response · Authors · 2021-11-20
**General Responses I**

**Dear AC and Reviewers,**

We are excited to see the level of appreciation expressed by all three reviewers and their consensus regarding our work's novelty and potential impact. Indeed, **Reviewer EL9b** states that "the whole paper is written in high-quality and a clear manner". According to **Reviewer A1LA**, “the proposed method is novel”, “the paper proposes an interesting idea” and “is potentially a good paper”. **Reviewer acZh** comments that our method is “a novel formulation of neural networks based on prior dynamical systems theory for an important machine learning problem” and “the application of it to the task of predicting model performance is novel as far as I can tell”.
We appreciate all the reviewers for their constructive and insightful comments on our paper. After addressing these comments, the manuscript has been further improved with better clarity. Below are our general responses to the issues regarding (I) overall structure and motivations, (II) purpose of random initialization, (III) readability of figures on performance, and the point-to-point responses to these comments, and the changes made in the revised manuscript are highlighted in blue. We are sincerely looking forward to communicating with the reviewers through OpenReview to address all your concerns.

**(I) Overall structure and motivations.**
> **Reviewer A1LA**: I do not find the relationship between the new layers and what is described in section 4.2 and 4.3.

> **Reviewer EL9b**: Weaknesses: Clarity. The paper uses multiple prior ideas from dynamical systems and machine learning literature but many of these choices are not well motivated or explained. This makes it difficult for a broad ML audience to understand the paper and its significance.

Here we provide a brief overview of our work to bring different pieces together. We view the NN training as a dynamical system, and directly model the evolution of the trainable weights in the SGD based training as a set of differential equations (**Section 3**), characterized by a general dynamics in Eq. 1. Usually,  it is convenient to study the dynamics of agents (trainable weights in our case) on a regular network, where each node represents an agent in the dynamical system and the interactions of agents are governed by Eq. 1. Many powerful techniques have been developed in network science and dynamical systems, e.g. the universal metric $\beta_{\rm eff}$ developed in [Gao et al. 2016] to quantify and categorize various types of networks, including the biological neural networks [Shu et al. 2021]. Because of the generality of the metric, we analyze how it looks on artificial neural networks which are designed to mimic the biological counterparts for general intelligence. Therefore, an analogue system of the trainable weights under the context of the general dynamics is set up in our framework. To the end, we build a line graph for the trainable weights (**Fig. 1a and Section 4.1**) and “rewrite” (**Section 4.2 and Appendix C**) the training dynamics in the form of Eq. 1, which includes a self-driving force $f(\cdot)$, an external driving force $g(\cdot,\cdot)$ and an adjacency matrix $P$ (**Eqs. 8 & 9 in Section 4.2**).

The reformulated training dynamics yields a simple yet powerful property. It is proved that as the neural network converges, $\beta_{\rm eff}$ approaches zero (**Theorem 1 in Section 4.3**). As shown in Fig. 1(c) and Section 4.3, we exploit the property to predict the final accuracy of a neural network model with a few observations during the early phase of the training, and apply it to select the pre-trained models (**Algorithm 1 in Section 4.4**). Generally speaking, the metric $\beta_{\rm eff}$ should be calculated for the entire neural network. However, because many state-of-the-art neural network models have large-scale trainable weights. If all layers are considered, it will be prohibitive to compute the associated $\beta_{\rm eff}$. We make a compromise, and estimate $\beta_{\rm eff}$ of the entire network using $\beta_{\rm eff}$ of the NCP unit (i.e., a partial of the entire network, see **the second to the last sentence of Section 4.3**). It’s confirmed from our empirical experiments (**Section 5**) that the simplified, lightweight version of $\beta_{\rm eff}$ is still effective in predicting the final accuracy of the entire neural network.

---

> ### Author Response · Authors · 2021-11-20
> **General Responses II**
>
> The metric $\beta_{\rm eff}$ developed by Gao et al. (2016) is universal to characterize different types of networks. **Although our framework utilizes the metric, our application to artificial neural network training dynamics and the related theoretical results as specified by Theorem 1 are novel**. Specifically, it is applied to study the NN training (**Section 3**) in our framework and predict the final accuracy of a neural network with a few observations during the early phase of the training (**Fig. 1c**). But $\beta_{\rm eff}$ relies on the adjacency matrix $P$ of $G_A$ (**Eq. 3**). To derive $P$, we resort to a reformation (**Section 4.2 and Appendix C**) of the training dynamics in the same form of the general dynamics (**Eq. 1**). One issue in calculating $\beta_{\rm eff}$ is its complexity if the entire $G_A$ is considered. As a resolution, we propose to use the lightweight $\beta_{\rm eff}$ of the NCP unit -- a partial of $G_A$ -- to predict the final accuracy of the entire network (**Sections 4.3 & 4.4**).
>
> We hope the above paragraphs are sufficient to explain our motivations and clarify the relationship between different sections.  The major changes in the revised version include:
>
> (1) In **Fig. 1**, we add an arrow from (b) to (a) and an arrow from (a) to (c) to highlight the dependency of $\beta_{\rm eff}$ on the adjacency matrix $P$, which is derived from the line graph $G_B$ of the NCP $G_A$ (Section 4.2). To get the associated line graph $G_B$,  the NCP $G_A$ needs to be passed through the network mapping (Section 4.1). Also, in the caption, we add two more sentences for better explanation of the relations.
>
> (2) In **Algorithm 1**, we add one more step following Step 3 to show how the weighted adjacency matrix $P$ is obtained.
> The text immediately following Section 4 is rewritten as a brief summary of the connections between the subsections for better readability.
>
> (3) **Appendix H** is added to cover the mean-field theory related derivations.
>
> (4) **Appendix I** is added to include the above explanation and a flowchart (Fig.I.7) of our framework’s main procedure.
>
> **References:**
>
> [Gao et al. 2016] Gao, Jianxi, Baruch Barzel, and Albert-László Barabási. "Universal resilience patterns in complex networks." Nature, 2016.
> [Shu et al. 2021] Shu, Pin, Hong Zhu, Wen Jin, Jie Zhou, Shanbao Tong, and Junfeng Sun. "The Resilience and Vulnerability of Human Brain Networks Across the Lifespan." IEEE Transactions on Neural Systems and Rehabilitation Engineering 29 (2021): 1756-1765.
>
> **(II) Purpose of random initialization.**
> > **Reviewer acZh**: By initializing it randomly and freezing it, a random amount of fixed noise is introduced into the process and it is unclear what the purpose of this is.
>
> > **Reviewer A1LA**: I'm not sure if I fully understand NCP units. Why are the weights randomly initialized and frozen during fine-tuning?
>
> We introduce the NCP unit to capture the training dynamics of the pre-trained model during fine-tuning, and to calculate $\beta_{\rm eff}$.  It serves as a passive measurement unit to predict the performance of neural network models with $\beta_{\rm eff}$.  To eliminate the intervention to the fine-tuning procedure of the transferred model (Fig. 1b), and reduce its impacts on the performance evaluation, as its name suggested, we limit it to the minimum function, i.e., receiving input signals from the bottom layers and making predictions. The randomly initialized and frozen NCP unit can run with no BP weights’ updates, having little impact on the training procedure of the bottom layers. We have also tested (1) a deterministic NCP unit with all-one initialization and (2) a trainable NCP unit. For all-one initialization, the network can not be trained at all; for trainable NCP, it can achieve better accuracy, but its ranking performance is less stable than the frozen NCP (drops as the length of learning curve increases, see the figure below for CIFAR10). More importantly,  the random initialization can make flexible evaluation on various types of architectures. “For full training of the best model, one can either retain or remove the NCP and fine-tune the selected model” (see the last sentence in Section 4.4).
>
> [Fig: Frozen vs Trainable NCP](https://imgur.com/a/s1mwcun)

---

> > ### Author Response · Authors · 2021-11-20
> > **General Responses III**
> >
> > **(III) Readability of figures on performance.**
> > > **Reviewer acZh**: The figures are small and difficult to read. A better presentation of the results such as a table could be added to the main paper while the figures are either added to the appendix or enlarged.
> >
> > > **Reviewer EL9b**: I am not 100% sure about the performance, because the author reports the accuracy with figures rather than tables.
> >
> > We regret that the reviewers could not fully examine the numerical performance based on the figures. We agree with the reviewers and greatly appreciate the suggestion to use tables instead. Following the suggestion, we (1) adjust the vertical space of the sub-figures and increase the font size of the notations in Fig. 2; (2) replace Fig. 4 with Table 1, and (3) put Fig. 4 to Appendix F (now Fig. F.5). Due to the page limit, it is hard to include all numerical results in Table 1. We just report the comparison results for learning curves of lengths 5 and 10. Moreover, to avoid any possible miscommunication, it will be highly appreciated if the reviewers can specify which figures are referred to as the sources of the concern. We are also happy to provide the detailed numerical results for other figures at the request of the reviewers.

---

### Author Response · Authors · 2021-11-22
**Looking forward to reviewer's feedback on our revised version and rebuttal**

Dear Area Chair and Reviewers,

Today (22th) is the deadline for allowing making further updates on our submission. While we are happy to continue to interact with the reviewers and area chair through 29th, we would like to ensure the current updated version is clear to reviewers and is helpful for addressing the reviewers' concerns and suggestions. We look forward to the post-rebuttal feedback and continued discussion from area chair and reviewers!

Sincerely,

Authors

---

> ### Author Response · Authors · 2021-11-29
> **Follow-up message from authors**
>
> Dear Area Chair and Reviewers,
>
> As the discussion deadline is closing soon, we would like to follow up to ensure we have successfully conveyed the merits and main contributions of our work. We took the silence of the post-rebuttal discussion as a positive sign indicating our revised version and responses had addressed your concerns. In the meantime, we are happy to answer any questions the AC and Reviewers may have. Please don't hesitate to let us know!
>
> Sincerely,
>
> Authors

---

### Decision · Program_Chairs · 2022-01-20

**Decision:**

Reject

**Comment:**

The paper proposes a method for inferring which of a set of pretrained neural networks, once fine-tuned on a transfer task, will generalize the best. This is accomplished by deriving a quantity based on a mean-field approximation of a dynamical system defined on the adjacency matrix of the weights of a neural network, known as the "neural capacitance". The model selection procedure involves attaching a fixed, randomly initialized network onto the outputs of the pretrained network and fine-tuning for a small number of iterations, and computing the metric; the fixed network is called the "neural capacitance probe" (NCP).

Reviews, though low confidence, awarded borderline scores, and a central concern was clarity and motivation, in particular the role of the NCP. acZh, the highest confidence and most verbose reviewer, echoed these concerns along with specific criticisms, for example about the heavy reliance on Gao et al (2016) without elaboration. The authors have responded in considerable depth but unfortunately the reviewer has not acknowledged these responses. On the NCP, the authors note that this is an approximation to the ideal metric that they have empirically validated.

Reading the updated draft, I find myself still concurring with reviewer acZh in large degree. The draft has improved with the noted additions, such as Appendix G devoted to an explanation of Gao et al (2016), but the presentation is still quite challenging to follow. I am left with fundamental questions about the soundness of the approximation being made, its wider applicability, and the many arbitrary decisions regarding the architecture of the NCP that appear out of nowhere. How sensitive is the procedure to these choices? Did the authors tune these architectural hyperparameters? Using what data? The table of results does not include units, and for a paper proposing a general purpose metric I'd ideally want to see a a robust rationale for hyperparameter selection of method-specific hyperparameters as well as a rigorous statistical treatment of the method's performance. Since it involves an approximation, a comparison to the "ideal" or "exact" procedure on a toy problem where the latter is feasible would strengthen the paper considerably. I do appreciate the breadth of architectures and datasets examined, but I believe the central focus of the paper should be explaining the mathematical motivation (perhaps at a higher level and deferring more detail to the appendix), why precisely it makes sense in the context of neural networks (also raised by acZh, with an answer provided that I believe partially addresses this) and justifying the concrete, approximate instantiation of the method involving the NCP and the hyperparameter selection and evaluation protocol that led you to the particular NCP employed.

At a higher level, this is a very mathematically dense paper that relies considerably on concepts outside of what might be considered typical expertise in the ICLR community, reflected in the confidence scores of the reviewers. While I feel that the issues described above already preclude acceptance at this time, I believe it may be difficult to do the proposed method justice in the short conference paper format, and would suggest to the authors to consider a journal submission instead, where a didactic presentation can be given the full attention it deserves without the difficulty created by length constraints.

Finally, I'd like to apologize to the authors for the non-responsiveness of the Area Chair. The original Area Chair was not able to complete their duty and I have been belatedly assigned this paper to evaluate it, and it is clear that not as much discussion took place as would have been ideal.